# LES study of the impact of moist thermals on the oxidative capacity of the atmosphere in southern West Africa

Fabien Brosse[1], Maud Leriche[1], Céline Mari[1], and Fleur Couvreux[2]

[1]Laboratoire d'Aérologie, Université de Toulouse, CNRS, UPS, France
[2]CNRM, Météo-France & CNRS, Toulouse, France

*Correspondence to:* Fabien Brosse (fabien.brosse@aero.obs-mip.fr)

**Abstract.** The hydroxyl radical (OH) is a highly reactive species and plays a key role in the oxidative capacity of the atmosphere. We explore the potential impact of a convective boundary layer on reconciling the calculation-measurement differences for OH reactivity (the inverse of OH lifetime) attributable to the segregation of OH and its reactants by thermals and the resulting modification of averaged reaction rates. The Large-Eddy simulation version of the Meso-NH model is used, coupled on-line with a detailed chemistry mechanism to simulate two contrasted biogenic and urban chemical regimes. In both environments, the top of the boundary layer is the region with the highest calculated segregation intensities but with the opposite sign. In the biogenic environment, the inhomogeneous mixing of isoprene and OH leads to a maximum decrease of 30% of the mean reaction rate in this zone. In the anthropogenic case, the effective rate constant for OH reacting with aldehydes is 16% higher than the averaged value. OH reactivity is always higher by 15 to 40% inside thermals in comparison to their surroundings as a function of the chemical environment and time of the day. Since thermals occupy a small fraction of the simulated domain, the impact of turbulent motions on domain-averaged total OH reactivity reaches a maximum decrease of 9% for the biogenic case and a maximum increase of 5% for the anthropogenic case. Accounting for the segregation of air masses by turbulent motions in regional and global models may increase OH reactivity in urban environments but lower OH reactivity in biogenic environments. In both cases, segregation alone is insufficient for resolving the underestimation between observed and modeled OH reactivity.

*Copyright statement.* TEXT

## 1 Introduction

The hydroxyl radical (OH) is an efficient cleansing molecule present in the troposphere. It is mainly produced during the daytime through the reaction of water vapor with $O(^1D)$, formed by ozone photolysis, while nitrogen oxides and volatile organic compounds (VOC) are its major sinks. OH is highly reactive and reacts with numerous chemical species, controlling their chemical lifetimes (Ehhalt, 1999). Both OH concentrations and its reactivity are key elements of the oxidative capacity of the atmosphere. Several field campaigns have been conducted to study total OH reactivity for urban and forested environments.

Measured reactivities have been compared to calculated reactivities obtained by summing OH reactant concentrations and multiplying them by their reaction rate constants. A missing part, corresponding to the difference between measured and calculated OH reactivity, is found not only under urban or biogenic conditions but also in clean remote regions. Measured OH reactivity in urban areas has been shown to be similar (less than 10%) to calculated OH reactivity in New York (Ren, 2003), in Houston (Mao et al., 2010) and in a controlled urban environment (Hansen et al., 2015). However, discrepancies in urban areas have been observed between measured and calculated OH reactivity in Nashville (Kovacs et al., 2003) (35% less for calculated reactivity), in Mexico (Shirley et al., 2006) (25%) and in Tokyo (Sadanaga, 2004) (25%). The differences between measured and calculated total OH reactivity are even higher in forested areas. Di Carlo (2004) found an unexplained fraction of 50% in measured OH reactivity during the PROPHET campaign in Michigan. These results are comparable to the missing part (50 to 58%) calculated from measurements made in a boreal forest in Finland (Sinha et al., 2010; Nölscher et al., 2012). Similarly, Nölscher et al. (2016) calculated an accounted fraction of measured OH reactivity close to 49% in an Amazonian rainforest.

As shown by Williams and Brune (2015), atmospheric models do not correctly simulate observed total OH reactivity. Attempts to use numerical models to explain the missing fraction of OH reactivity have proved to be insufficient. Indeed, Edwards et al. (2013) found an underestimation of 30% of OH reactivity in a box model with a detailed chemical mechanism for the OP3 project. In the PRIDE-PRD campaign, Lou et al. (2010) found discrepancies of $+/-$ 10% between the results of the OH reactivity model and the measurements, also using a box model. Similarly, Mogensen et al. (2011) used a column model to elucidate the missing part of OH reactivity, but only explained 30 to 50% of the OH reactivity measured over a forest in Finland. Chatani et al. (2009) used a three-dimensional model with coarse resolution to fill the gap in OH reactivity but 40% of the measured OH sinks remained unexplained. The difficulty of getting models to represent OH reactivity could be partly due to as yet non-discovered OH reaction pathways, and which are therefore not implemented in atmospheric models. OH recycling by the isoprene oxidation chain in forest environments characterized by low $NO_x$ (sum of NO and $NO_2$) conditions (i.e. <1 ppb) was proposed to explain the uncertainties in the simulated $HO_x$ (sum of OH and $HO_2$) budget (Lelieveld et al., 2008; Butler et al., 2008; Peeters et al., 2009; Pugh et al., 2010; Stone et al., 2011). However, Stone et al. (2011) studied several proposed OH recycling mechanisms present in the literature and found that biases from OH and $HO_2$ concentrations still exist whatever the mechanism. One possible issue in total OH reactivity retrieval not mentioned by previous studies could lie in neglecting turbulent motions in the transport of chemical compounds in the boundary layer. Indeed, turbulence can spatially segregate or bring together chemical species, reducing or increasing the mean reaction rate and thus chemical reactivity. However, as far as we know this physical process has not been investigated in previous studies. The time response of current OH reactivity measurement techniques is not yet sufficient to directly resolve the smallest relevant turbulent spatial scales. The limitation in time resolutions range from 30 seconds for LIF-based methods (Kovacs and Brune, 2001; Sadanaga, 2004) to one minute for the CRM method (Sinha et al., 2008). In comparison, Pugh et al. (2011) and Dlugi et al. (2010) used direct isoprene and OH concentrations measurements with temporal resolution of a few seconds, fast enough to estimate the segregation of the compounds.

The atmospheric boundary layer has a turbulent structure characterized by strong and narrow updrafts surrounded by weak and large descending areas (Molemaker and Vilà-Guerau de Arellano, 1998; Schumann, 1989). Wyngaard and Brost (1984)

considered passive scalars and found that surface bottom-up transport plays a more important role in the vertical diffusion than the entrainment zone top-down transport in a convective boundary layer. Updrafts in the boundary layer lead to the spatial discrimination of pollutant concentrations between thermals and their environment. This heterogeneity in chemical species redistribution influences the mean reaction rate obtained when considering averaged reactant concentrations (Vilà-Guerau de

Arellano and Cuijpers, 2000). Using an idealized simulation, Molemaker and Vilà-Guerau de Arellano (1998) showed that, for a second-order reaction implying a top-down and a bottom-up species, reaction rates are maximum in updrafts near the surface and in downdrafts at the top of the boundary-layer. Segregation between VOC and the OH radical was first addressed by the numerical study of Krol et al. (2000) who investigated the turbulence effects on the mean reaction rates of these species. Ouwersloot et al. (2011) studied the inefficiency of turbulent mixing over heterogeneous surfaces and found that the highest

reaction rates for isoprene and OH are located in thermals at the top of the boundary layer. The chemical reactivity of the boundary layer is therefore determined by the capacity of turbulence to mix reactive species (Molemaker and Vilà-Guerau de Arellano, 1998). However, modelling and experimental studies investigating heterogeneities in the boundary layer have focused on OH radical concentrations rather than on OH reactivity due to the instrumental limitations discussed above.

Vertical motions associated with clouds and sea-breeze impact the atmospheric chemistry and pollution levels near the

surface since they dilute chemical species and increase the upward transport of surface emissions (Chen et al., 2012). In the case of updrafts leading to cloud formation, Vilà-Guerau de Arellano et al. (2005) found a decrease of 10 to 50% of tracer mixing ratios averaged over the boundary-layer with respect to a situation without clouds due to the deepening of the boundary-layer. Clouds have multiple impacts on the atmospheric boundary layer as they induce turbulent mixing of chemical compounds which modifies reaction rates. They also modify incoming solar radiation, which in turn disturbs photolysis reactions and alters

the emissions of biogenic compounds, such as isoprene. Finally, they alter atmospheric chemistry due to soluble gas washout and chemical reactions occurring in cloud droplets.

Cloud cover over West Africa is an important feature of the African monsoon but is poorly represented by global models (Knippertz et al., 2011; Hannak et al., 2017). This could lead to overly low simulated clouds and overly high incoming radiation at the surface, implying excessively high diurnal temperature and relative humidity cycles over this region. The nocturnal low-

level stratus was studied during the monsoon period at Parakou (Benin) by Schrage et al. (2007) with radiosondes. The authors found that turbulent processes are responsible for cloudy nights whereas clear nights are associated with a nocturnal inversion leading to the decoupling of the surface and lower atmosphere. Schrage and Fink (2012) investigated nighttime cloud formation. They observed that the presence of a nighttime low-level jet induces the shear-driven vertical mixing of moisture accumulated near the surface. This leads to stratus formation whose cover is likely to persist until the early afternoon when it breaks up

to form cumulus clouds (Schrage et al., 2007; Schrage and Fink, 2012). However, studying the impact of this specific cloudy environment on the turbulent transport of chemical species in tropical West Africa has not been reported.

High resolution simulations which explicitly resolve the turbulent and convective advection terms were conducted (Vilà-Guerau de Arellano and Cuijpers, 2000; Vilà-Guerau de Arellano et al., 2005; Ouwersloot et al., 2011; Kim et al., 2012, 2016) to assess the impacts of clouds and the convective boundary layer on the mixing of chemical compounds. However, previous

numerical studies on the impact of the turbulent mixing of chemical compounds mainly used relatively simple or only slightly

more detailed chemical schemes (e.g., Vilà-Guerau de Arellano and Cuijpers (2000),Vilà-Guerau de Arellano et al. (2005) and Ouwersloot et al. (2011)), resulting in possible limitations in the representation of the atmospheric chemistry. Besides, more recent studies by Kim et al. (2012) and Kim et al. (2016) used a more detailed chemical scheme derived from Mozart v2.2, allowing the formation of OH radicals initiated by peroxy radicals. This limits the spatial heterogeneities caused by the reactions consuming the OH radical.

The goal of this work is to evaluate the role of thermals on OH reactivity in the framework of a convective boundary layer with contrasted chemical environments in southern West Africa. It focuses in particular on investigating turbulence as a possible explanation of the discrepancies between calculated-measured OH reactivities mentioned in the literature. Two contrasted chemical regimes represented by a detailed chemical scheme are studied by using Large-Eddy Simulations. The first simulation is influenced by biogenic emissions whereas the second is characterized by anthropogenic emissions representative of Cotonou (Benin) in the center of the domain. Based on a conditional sampling implemented in the model, the thermals are discriminated in these simulations, allowing the specific chemical regime inside thermals to be investigated. The model experiments are presented in section 2. Section 3 presents the dynamical and chemical results for the two cases while section 4 presents the discussion relating to these results.

## 2  Simulation description

### 2.1  Model configuration

LES simulations are performed with the mesoscale non-hydrostatic atmospheric model Meso-NH (http://mesonh.aero.obs-mip.fr/mesonh/) version 5.2.1, developed jointly by the Laboratoire d'Aérologie and the Centre National de le Recherche Météorologique (Lac et al., 2018). Cloud microphysical processes are represented by the ICE3 scheme (Pinty and Jabouille, 1998) that includes six different types of hydrometeors. The turbulence is solved by a 3-dimensional scheme using a prognostic equation for the turbulent kinetic energy (Cuxart et al., 2000) with the turbulent mixing length given by the mesh size. Surface processes and interactions with the atmosphere are simulated by the SURFEx model (Masson et al., 2013) coupled with Meso-NH.

The resolution used is 50m × 50m for a domain size of 10 km × 10 km (200 × 200 grid points). 10 km is the targeted mesh size of an increasing number of current large-scale chemistry models. Along the vertical, 121 levels are stretched from $\Delta z =$ 20 m at the surface to 250 m on top of the domain at 20 km and the boundary conditions are cyclic.

The simulation is run for three days in which two days consist of the spin-up for chemistry. The same dynamical conditions and the same initial and forcing thermodynamical fields are used for each day. The results are shown only for the third day and from 0600 to 1700 UTC (LT=UTC+1), when the convective boundary-layer is well developed. A passive scalar is emitted only during this part of the simulation with a constant emission rate and is used to determine the boundary layer height (see section 2.4.1). The thermals are identified by the conditional sampling method implemented in the model by Couvreux et al. (2010), which relies on a first-order decay passive tracer mixing ratio emitted with a constant flux at the surface. In brief, in order to be considered as thermals, air parcels at a given altitude $z$ must satisfy simultaneous conditions such as a positive vertical

velocity anomaly $w' > 0$ and tracer anomalies $sv'(z)$ greater than the standard deviation of the tracer concentration $\sigma_{sv}(z)$ and a minimum threshold $\sigma_{min}(z) = (0.05/z) \int_0^z \sigma_{sv}(k)dk$. In the cloud layer, a supplementary condition is that the grid box has to be cloudy. This passive tracer is emitted from the beginning of the third day of simulation.

The initial and forcing dynamical fields are taken from Couvreux et al. (2014) who used a single column model to study the representation of the diurnal cycles of meteorological parameters at four observation sites in West Africa (Agadez and Niamey in Niger, Parakou and Cotonou in Benin). Here we focus on the "cloudy" regime of Couvreux et al. (2014), representative of the climate encountered close to the Gulf of Guinea. The vegetation present in our simulation is dominated by tropical crops and open shrublands (35% of the domain for each type), sea (15%), inland water (5%), wetlands (5%) and tropical grasslands (5%) and a high moisture content is prescribed with soil water indexes of 0.7 and 0.74 for the surface and the ground, respectively. The initial conditions and composite large-scale advections were extracted from the ECMWF re-analysis (Agustí-Panareda et al., 2010) prepared for the AMMA campaign (Redelsperger et al., 2006). The present simulation was initialized at 0600 UTC with stable initial conditions extracted from the ECMWF AMMA reanalysis (black curve in Fig. 1b). The large-scale conditions from the ECMWF re-analysis are weak in magnitude but includes sea-breeze circulations from the surface to 500 m, linked to moist and cool advection throughout the simulation, topped by the advection of dry and warm air from 1000 to 3000 m.

## 2.2 Chemical model setup

The chemical scheme ReLACS 3.0 (Reduced Lumped Atmospheric Chemical Scheme version 3.0) (Tulet et al., 2006) applied is a reduced version of the Caltech Atmospheric Chemistry Mechanism (CACM) (Griffin, 2002). This mechanism describes the reaction system of gaseous ozone precursors as well as of Secondary Organic Aerosols (SOA) with 365 reactions involving 87 species.

For both simulations, the initial vertical profiles of the main primary chemical species are taken from airborne measurements made during the B235 flight of the AMMA campaign performed by the BAE-146 aircraft. (Table 1). This particular flight gives access to measurements performed in the boundary layer over a tropical forest in the north of Benin ($10.13^oN$, $2.69^oE$) during the early afternoon (Stone et al., 2010).

Biogenic emissions (Table 2) are taken from the MEGAN-MACC (Model of Emissions of Gases and Aerosols from Nature - Monitoring Atmospheric Composition and Climate) inventory (Sindelarova et al., 2014) except for NO, which is not available in this inventory. Biogenic $NO_x$ emissions from the GEIAv1 (Global Emission InitiAtive) inventory (Yienger and Levy, 1995) proved to be too low for the region studied in comparison to estimations based on airborne measurements during the AMMA campaign (Stewart et al., 2008; Delon et al., 2010). Therefore, a maximum value of 10 ng.N.m$^{-2}$.s$^{-1}$ was set for nitrogen oxide emissions from soils in the simulation.

The emissions are constant in space and time except for NO, isoprene and monoterpenes, for which a Gaussian diurnal cycle is used. For biogenic NO, the maximum emission occurs at 1300 UTC and the standard deviation of the Gaussian curve is equal to 3 hours. These parameters are set in order to approximate the ground temperature since Mamtimin et al. (2016) and Yienger and Levy (1995) noted that NO emissions from soils are closely linked to soil temperature. For isoprene and the sum

of monoterpenes (represented as ISOP, BIOL and BIOH in the chemical scheme), the maximum occurs at 1200 UTC and the standard deviation is equal to 2.5 hours, chosen to fit the diurnal evolution of the solar radiation reaching the surface. This is in agreement with Guenther et al. (1991), who showed that isoprene emissions are thought to be null when Photosynthetically Active Radiation (PAR) is equal to zero and maximum when the PAR exceeds the value of 1000 $\mu$mol.m$^{-2}$.s$-1$. For isoprene

and monoterpenes, the maximum emission values were defined to ensure that an equal amount of chemical species is emitted over one day compared to the constant value provided by MEGAN-MACC.

The anthropogenic emissions are provided by a squared patch in the center of the domain. Its area is chosen as equal to half the domain area. However, the cyclic boundary conditions applied for these simulations tend to homogenize the chemical mixing ratios. This especially affects long-lived species in the boundary layer and leads to the deletion of the biogenic emission

signature. The values of anthropogenic emissions are representative of Cotonou (Table 2) (Junker and Liousse, 2008).

In the following, the study focuses on isoprene for the biogenic case because it is the major biogenic VOC emitted into the atmosphere and influences ozone and secondary organic aerosol formation (Guenther et al., 2006). For the anthropogenic case, attention is given to the lumped C>2 aldehydes (ALD2 in the chemical scheme) because they have both primary and secondary sources and their role is very important in the troposphere as they contribute to the production of radicals and are precursors of

ozone (Williams et al., 1996).

## 2.3 Metrics

In order to study the competition between chemical reactivity and turbulent mixing, and inhomogeneity in chemical species mixing ratios, Schumann (1989) introduced two dimensionless numbers: the Damkhöler number and the segregation coefficient. The first corresponds to the ratio between the characteristic turbulence timescale $\tau_{turb}$ and the chemical reactivity

timescale $\tau_{chem}$. For a given compound A, the Damkhöler number $D_a$ is :

$$D_a(A) = \frac{\tau_{turb}}{\tau_{chem}(A)} \text{ with } \tau_{turb} = \frac{w^*}{h} \text{ and } \tau_{chem}(A) = \frac{r_A}{\sum sinks(A)} \tag{1}$$

Where $w^*$ and $h$ refer to the convective velocity and the boundary-layer height, $r_A$ is the mixing ratio of A and $\sum sinks(A)$ corresponds to the total chemical loss rate of A. The convective velocity is computed with $g$, the standard acceleration due to gravity, and $\theta$, the potential temperature, according to the relation $w^* = (\frac{g}{\theta} \cdot (\overline{w'\theta'_{v,s}}) \cdot h)^{1/3}$ where $\overline{w'\theta'_{v,s}}$ stands for the

buoyancy flux at the surface. Schumann (1989) distinguished different chemical regimes for the reaction between nitrogen oxide and ozone and found that the impact of turbulence on this reaction rate is highest for D$_a$ > 0.1. Later studies (Molemaker and Vilà-Guerau de Arellano, 1998; Vilà-Guerau de Arellano and Cuijpers, 2000; Vilà-Guerau de Arellano et al., 2005) have shown that the impacts of turbulence on atmospheric chemistry are expected to be maximum when D$_a \geq 1$. Therefore, this value will be used in the following to discriminate slow and fast chemical reactions in the boundary layer. If $D_a < 1$, then the

turbulent mixing is more efficient than the chemistry. If $D_a \simeq 1$, strong competition can be expected between dynamics and chemical reactivity. Finally, if $D_a > 1$, the chemical reactions occur faster than turbulent mixing.

The turbulent mixing causes fluctuations of chemical species mixing ratios in the LES domain, which can be quantified by the intensity of segregation. For a second order reaction involving two species A and B with a reaction constant $k$: $A + B \rightarrow C$, the intensity of segregation $I_S(A,B)$ is defined as:

$$I_S(A,B) = \frac{\overline{a'b'}}{\overline{a}.\overline{b}} \tag{2}$$

5 The lower case letters represent species mixing ratios at a grid point. The overbar denotes a spatial average, and the prime a deviation from this average. If $I_S(A,B) = -1$, then the two species are completely segregated and no reaction will take place between them. If $I_S(A,B) = 0$, the compounds are perfectly mixed. A positive segregation coefficient means that the covariance between species is similar and thus the mean chemical reaction rate would be higher in comparison to perfect mixing. The segregation is calculated with mixing ratio anomalies related to spatial averages in numerical models. Ouwersloot

10 et al. (2011) stated that these averages should correspond to the complete mixing volume in order to allow comparisons between models and measurements. In the following, the large-scale spatial average is calculated over the 10 km x 10 km model domain. An effective average reaction rate $R_e$ can be defined that includes the impact of turbulent mixing on chemical reactivity as:

$$R_e = k_e \cdot \overline{a} \cdot \overline{b} \text{ with } k_e = k \cdot (1 + I_S(A,B)) \tag{3}$$

Here, $k_e$ is the effective mean reaction constant. In the current LES experiments, chemical reaction rates are calculated with

15 a focus on OH. OH radical reactivity $R_{OH}$ corresponds to the inverse of OH lifetime $\tau_{OH}$ and is defined as:

$$R_{OH} = \frac{1}{\tau_{OH}} = \sum_i^n k_{(\chi_i + OH)} \cdot \chi_i \tag{4}$$

In Equation 4, $k_{(\chi_i + OH)}$ represents the reaction constant between OH and the $i$-th reactant, and $\chi_i$ corresponds to its concentration. As with the effective reaction constant $R_e$, effective reactivity for the OH radical $R_{OH}^e$ is defined by including the effect of turbulent mixing in Equation 4 as:

20 $$R_{OH}^e = \sum_i^n k_{(\chi_i + OH)}^e \cdot \chi_i \tag{5}$$

$$= \sum_i^n k_{(\chi_i + OH)} \cdot (1 + I_S(OH, \chi_i)) \cdot \chi_i \tag{6}$$

$$R_{OH}^e = R_{OH} + \sum_i^n k_{(\chi_i + OH)} \cdot I_S(OH, \chi_i) \cdot \chi_i \tag{7}$$

$R_{OH}$ denotes the OH total reactivity calculated with averaged values. In order to obtain the relative deviation of the total OH reactivity from the reactivity computed with averaged mixing ratios, factorization is performed on Equation 7, which results in:

$$R_{OH}^e = R_{OH}.(1 + \frac{\sum\limits_{i}^{n} k_{(\chi_i+OH)} \cdot I_S(OH, \chi_i) \cdot \chi_i}{R_{OH}}) \tag{8}$$

$$R_{OH}^e = R_{OH}.(1 + E_{R_{OH}}) \tag{9}$$

From Equation 9 the mean relative error, $E_{R_{OH}}$, found on total OH reactivity considering only averaged values is :

$$E_{R_{OH}} = \frac{\sum\limits_{i}^{n} k_{(\chi_i+OH)} \cdot I_S(OH, \chi_i) \cdot \chi_i}{R_{OH}} \tag{10}$$

The segregation intensity used to compute the mean error corresponds to the deviation from the averaged boundary layer values. This error on OH reactivity was not considered in previous numerical studies focused on identifying the missing part of OH reactivity. Indeed, using a box model or a single column model like Mogensen et al. (2011), Whalley et al. (2011) or Whalley et al. (2016) leads to neglecting the turbulent motions that could affect the redistribution of chemical species within the atmospheric boundary layer. This may imply an underestimation or an overestimation of OH reactivity as a function of the sign of $E_{R_{OH}}$. If $E_{R_{OH}}$ is positive or negative, then the effective OH reactivity $R_{OH}^e$ is either higher or lower, respectively, than the OH reactivity $R_{OH}$ found by neglecting the turbulent motions. Due to the crucial aspect of the OH radical in the atmosphere, this could subsequently modify the lifetimes of gaseous OH reactants such as ozone and carbon monoxide.

### 2.4 Simulation assessment

#### 2.4.1 Dynamics

The diurnal evolution of the boundary-layer height (BLH) is analyzed in Fig. 1a. It is diagnosed using two different methods. The first corresponds to a determination according to the bulk Richardson number method as presented in Zhang et al. (2014). The boundary layer height is defined as a threshold value for the bulk Richardson number $Ri_b$, computed at a given height with the virtual potential temperature $\theta_v$ and horizontal wind speeds $u_z$ and $v_z$ at this altitude and at the surface:

$$Ri_b = \frac{(g/\theta_{v0})(\theta_{vz} - \theta_{v0})z}{u_z^2 + v_z^2} \tag{11}$$

A clear diurnal cycle is observed, with the maximum height at 1400 UTC and the minimum during nighttime (Fig. 1a, red line). This first diagnostic did not include the cloud layer on the boundary layer. Daytime BLH is consistent with observations from Cotonou in West Africa studied by Gounou et al. (2012), who noticed daily variabilities ranging from 400 to 600 m

(Fig. 1a, black dots). In Gounou et al. (2012), the boundary-layer height is derived from radio soundings by comparing the virtual potential temperature at one level and the averaged value below. The slight differences between the simulated BLH and observations reveal the same biases noticed in the 1D-simulation performed by Couvreux et al. (2014). This may be explained by uncertainties on the derivations of large-scale advection fields.

The second method determines the boundary layer height according to Vilà-Guerau de Arellano et al. (2005) by identifying the height at which a passive bottom-up scalar emitted at the surface is equal to 0.5% of its surface value. Kim et al. (2012) mentioned that this definition could be valuable when studying boundary layer deepening due to clouds. This diagnostic captured the growth of the boundary layer height due to the development of cumulus clouds (Fig. 1a, blue line) and is used to mark the BL height in the following. The tracer used for this diagnostic was emitted only during the period of interest, from 
0600 to 1800 UTC.

  The range of simulated virtual potential temperature (Fig. 1b) overestimates the AMMA observations in Cotonou of Gounou et al. (2012) in the lowest 500m, as shown by Couvreux et al. (2014). At 0600 UTC, the model has a cold bias of -2K turning throughout the simulation to a simulated potential temperature overestimated by +2K at the end of the simulation. Both the dry and cloudy layers are identified by two inflections on the vertical profiles of the virtual potential temperature (Fig. 1b). The 
first inflection increases from 400 m at 0600 UTC to 600 m at 1700 UTC, corresponding to a thin inversion zone between the well-mixed dry layer and the cloudy layer above it. The second inflection defines the top of the boundary layer ranging from 500 m at 0600 UTC to 1600 m at 1700 UTC.

  The southern part of West Africa is a region characterized by high diurnal variability in cloud occurrence. Low-level stratus clouds form during the night and persist in the morning, these stratus then break up during the afternoon into cumulus clouds 
(Schrage et al., 2007). This feature is simulated in this work as low level stratus clouds that occupy a large fraction of the simulated domain in the morning (Fig. 2a and c). In the afternoon, the cloud deck breaks up and less uniform but higher cumulus clouds are simulated (Fig. 2b and d).

  Throughout the growth of the boundary layer, dry thermals develop capped by the temperature inversion zone. Some thermals penetrate this inversion (Fig. 2) and cloud formation occurs at the upper part of the updrafts. The vertical profiles of the fraction 
area (Fig. 3) occupied by thermals exhibit a local maximum at the height corresponding to the separation between the two layers. This altitude is variable through the simulation but tends to stabilize between 500 and 600m in the early afternoon and is associated with a preferential detrainment zone. The peaks observed at the surface and at the top of the cloudy layer correspond to two other local maxima. Due to the deepening of the boundary-layer, the top of the cloudy layer increases throughout the simulation (Fig. 2,3). Simulated convective velocity $w^*$ ranges from $1 cm.s^{-1}$ in the morning to $1 m.s^{-1}$ at 
midday, and the turbulence characteristic timescale $\tau_{turb}$ from $6h$ in the morning to $20\ min$ at midday.

### 2.4.2 Atmospheric chemistry

Typical diurnal cycles are obtained for isoprene in the biogenic case and for OH in the biogenic and anthropogenic cases in which both compounds exhibit maximum mixing ratios around midday (Fig. 4). For the biogenic case, the simulated isoprene mixing ratios averaged from the surface to 600 m reach a maximum close to 1 ppbv (Fig. 4a) at noon, when isoprene emissions

are highest. This is in the same range as the AMMA measurement studied by Saxton et al. (2007), who found a maximum of 1.5 ppbv on the composite diurnal cycle of isoprene at midday. In our study, the ozone mixing ratios did not exhibit strong variability throughout the day. The simulated values around 18 ppbv were lower than the observations recorded over forested areas (Table 3), ranging from 22 to 30 ppbv. The $NO_x$ mixing ratios were close to 0.2 ppbv on average around midday (Fig. 4b) and are in agreement with AMMA measurements where a mean value of 0.1 ppbv of $NO_x$ was observed. Simulated OH mixing ratios varied between 0 and 0.18 pptv (Fig. 4a), within the observational range of 0.05 to 0.15 pptv during the AMMA campaign (Table 3). The biogenic environment is characteristic of a limited $NO_x$ regime.

The chemical regime induced by the anthropogenic emissions is contrasted with the previous biogenic case (Fig. 4c and d), including $NO_x$ emissions nearly forty times higher at 1300 UTC. The averaged $NO_x$ mixing ratio was 1.5 ppbv at midday (Fig. 4d), which is similar to the averaged value of 1 ppbv measured during a flight over Cotonou during the AMMA campaign (Table 3). The increase in NO and $NO_2$ led to considerable oxidant formations. OH mixing ratios varied between 0 and 0.40 pptv in the simulation, in agreement with the AMMA measurements ranging from 0 to 0.50 pptv over Lagos (Table 3). Ozone was produced throughout the simulation but did not exhibit strong spatial variabilities. Its mixing ratios varied from 17 ppbv in the morning to 82 ppbv in late afternoon. This was higher than the observations recorded over several cities in West Africa, varying between 26 and 40 ppbv (Table 3). ALD2 increased continuously during this case through chemical production and emissions (Table 2). Its mixing ratios ranged from less than 1 ppbv in the morning to 11 ppbv at the end of the simulation. Unfortunately, no aldehyde observations, except formaldehyde, were available from the AMMA experiment despite the use of a PTR-MS due to interferences during the measurements for $m/z$=45 identified as acetaldehyde (Murphy et al., 2010).

A minimum of $NO_x$ was found around 1200 UTC for both cases and can be explained by two factors. The first is dynamical and is linked to the boundary layer. In the middle of the simulation, the boundary layer growth was maximal and induced dilution in a larger mixing volume. The second factor was chemical because at that instant, NO was efficiently converted into $NO_2$. However, $NO_2$ chemical transformations in a reservoir such as $HNO_3$, $HNO_4$ or PANs are net sinks for $NO_2$. The chemical balance between reservoir species and $NO_2$ represented 2.12% of the net destruction of $NO_2$ averaged over the domain at 20 m and 1200 UTC for the biogenic case, and 34.2% for the anthropogenic case. For both cases, the main reservoirs of NO2 were PAN1 and PAN2. Therefore, less $NO_2$ was available for conversion into NO, which explains the low NO mixing ratios at midday (Fig. 4b and d).

## 3  Vertical transport and chemical reactions in the convective boundary layer

### 3.1  Impact of turbulent mixing on the OH reactions

#### 3.1.1  Vertical transport and Damkhöler numbers

Isoprene is highly reactive, especially towards OH, and is rapidly consumed in the boundary layer linked to its chemical lifetime $\tau_{ch}$ approximately equal to 30 min in the biogenic case. The Damkhöler number of isoprene is close to 1 (Table 4), indicating that the isoprene chemical lifetime is comparable with the timescale of turbulent mixing. This implies that chemical

reactions and turbulent mixing are competing processes for this compound. Isoprene is both transported and consumed inside the thermals and exhibits vertical and horizontal gradients (Fig. 5a). The average profiles of isoprene decrease with altitude (Fig. 5a). The lower mixing ratios in the updraft-free region (dashed line) are close to the domain averaged values due to domain coverage by thermals (Fig. 3). Updraft regions contain higher isoprene mixing ratios with significant anomalies over the whole boundary layer, although decreasing with altitude.

For the biogenic case, OH has a very short chemical lifetime of the order of 0.2 s. The OH radical rapidly reaches steady state and is relatively undisturbed by the turbulent mixing, as evidenced by its large Damköhler number (Table 4). Therefore, OH mixing ratios (Fig. 5b) have almost homogeneous values below 600 m with no distinction between thermals and their environment. Average OH profiles (Fig. 5b) show no strong variability below 600 m in the boundary layer. Above this height, the thermals have lower OH concentrations on average with relatively small differences. The differences between thermals and their surroundings are more pronounced in the cloudy layer because clouds arise from air parcels transported by thermals and characterized by lower OH mixing ratios than the rest of the domain. Since OH is almost constant over the boundary layer, the highest reaction rate for the oxidation of isoprene by OH ($k_{OH+ISOP}[ISOP][OH]$) is located where the isoprene mixing ratios are the highest. This implies that both the surface and air lifted by thermals are preferential reaction zones in the boundary layer.

For the anthropogenic simulation, the reaction rate of ALD2 with OH is lower than the reaction rate of OH with isoprene. Its calculated Damköhler number ($\approx 0.17$) indicates that turbulent mixing dominates over chemical reactions (Table 4). Consequently, ALD2 is efficiently transported by updrafts (Fig. 5c). The contrast between concentrated air parcels lifted upwards by thermals and relatively diluted air outside is illustrated by the large differences in the ALD2 average profile in thermals and in the environment (Fig. 5c). As with isoprene in the biogenic case, the two-layer structure presented above is present on the vertical profiles. It results from the mixing in the sub-cloud layer and from the mixing within clouds.

OH mixing ratios are nearly twice as high in the anthropogenic environment as in the biogenic case, inducing a more reactive atmosphere and a shorter chemical lifetime for species whose OH is the main sink in the boundary layer. OH has a very short chemical lifetime of 0.07s in this simulation, which represents a large Damköhler number close to 12 000. OH mixing ratios are maximal in thermals (Fig. 5d) due to the transport of OH precursors, such as $NO_x$, and fast steady state. Conversely, lower mixing ratios in average OH profiles (Fig. 5d) correspond to regions without updrafts, leading to strong OH anomalies within updrafts from the surface to the top of clouds.

### 3.1.2  Vertical profile of segregation intensity

Negative values of segregation coefficients up to -30% are calculated at the top of the cloudy boundary layer from 1000 to 1700 UTC which means that OH and isoprene are partly segregated in this frontier zone. In other words, the hypothesis of a well-mixed atmosphere would lead to a 30% overestimation of the reaction rate at the frontier between the boundary layer and the free troposphere. The negative segregation (Eq. 2) means the anomalies of isoprene and OH have opposite signs, as shown in figure 5a. This is due to lower OH mixing ratios in thermals than in the environment (Fig. 5b). These results are consistent with the previous studies of Li et al. (2016); Kim et al. (2016); Ouwersloot et al. (2011) (see Discussion).

In the biogenic case, isoprene anomalies in thermals are considerable from the surface (+0.48 ppbv on average at midday) to the top of the boundary layer (+0.1 ppbv on average at midday) and are thought to be always positive as OH is uniformly emitted at the ground (Fig. 5a). On the contrary, OH mixing ratios are almost constant in the boundary layer at 1200 UTC (Fig. 5b), so the magnitude of OH anomalies are expected to be low (-0.03 pptv on average at midday at the top of the boundary layer). Besides, due to its very short chemical lifetime, OH quickly reaches equilibrium with its surroundings, implying that its fluctuations are mostly due to thermals transporting air originating from different chemical environments. Thus, isoprene anomalies are thought to be the major driver of the magnitude of segregation over the boundary layer whereas changes in OH anomalies are related to changes in the segregation sign.

Positive values around +5% are calculated at 700 meters starting from 1400 to 1800 UTC (Fig. 6a). The intensity of segregation becomes positive due to positive anomalies of both compounds. Due to decrease in isoprene emissions in the afternoon, OH destruction slows down, especially inside thermals. They are still active in transporting enough NO to react with $HO_2$ to produce OH, inducing higher OH mixing ratios inside updrafts than in the surroundings (+0.02 pptv on average at 1600 UTC).

Before 0900 UTC near the surface, the segregation coefficient in the anthropogenic simulation between OH and ALD2 is negative up to -8% in the lower 200m (Fig. 6b), due to the anthropogenic emission patch. As chemical equilibrium is not yet reached, more of the OH radical is destroyed through its reaction with recently emitted compounds than that which is produced (not shown). This means that OH is less concentrated inside updrafts at that moment so its anomalies are negative near the surface. Simultaneously, positive segregations develop at the top of the boundary layer from 0700 UTC to 1230 UTC with a maximum of 16% at 1000 UTC and from 1530 to 1730 UTC. The positive segregation is related to the concomitant transport of ALD2 and precursors of OH by thermals. Moreover, the high segregation values correspond to the presence of clouds between 0.6 and 0.9 km (Fig. 2c), simultaneously with a high cloud fraction over the domain (>0.6) (Fig. 6b). This specific point is discussed in the discussion section. As ALD2 is emitted at the surface, its anomalies are high and positive inside updrafts. For example, at midday, anomalies are +0.5 ppbv on average at the surface and nearly +4 ppbv at the top of the boundary layer. However, in this case, OH anomalies are more difficult to predict due to the spatial heterogeneities of chemical emissions. Local changes in OH production and destruction explain the changes in the segregation sign throughout the simulation. Except for positive segregation simulated between 500 and 1400 m from 1130 to 1600 UTC with values ranging from 2 to 4%, ALD2 and OH can be considered well-mixed in the central part of the boundary layer. The ALD2 oxidation reaction by OH is accelerated up to 16% at the top of the cloudy layer from the morning to the early afternoon compared to a perfect mixing assumption.

Regarding the two simulations, the segregation has both spatial and temporal variations. The maximum values of the segregation coefficient are calculated near the top of the boundary layer. Below and during daytime, the considered species are well-mixed for both cases. This means that in the biogenic environment, the highest decrease induced by the thermals of isoprene + OH reaction is located near the top of the boundary layer. It also implies that in the anthropogenic environment, the highest increase induced by the thermals of isoprene and ALD2 + OH reaction is also located near the top of the boundary layer.

## 3.2 OH budget and reactivity in a convective boundary layer

The previous part emphasized the non-uniform mixing between isoprene and OH for the biogenic case and between OH and ALD2 for the urban case, and the modification of the reactions rates between these species. However, this feature must be taken into account for every OH reactant in order to obtain the full picture of total OH reactivity and gain insight into how the
Meso-NH model computes the OH budget in different chemical regimes.

### 3.2.1 OH budget in thermals versus surroundings

In order to identify and quantify the major OH sources and sinks in the boundary layer, the instantaneous chemical budget of OH at 20 meters above ground level is investigated at 1200 UTC for both environments (Fig. 7). This height is the first level in the model and computing the chemical budget at this height leads to uncertainties due to subgrid-scale mixing and
chemistry (Vinuesa and Vilà-Guerau de Arellano, 2005). However, it makes it possible to compare the model results with the measurements in the literature.

The budget distinguishes between updraft and updraft-free columns. Percentages are related to the fraction of the overall production and destruction within or outside thermals. In the biogenic case (Fig. 7a), the OH budget is dominated by its destruction by isoprene oxidation (41.7% of the total loss in updrafts and 29.3% of the total loss in the rest of the domain) and
its production by the reaction of peroxy radicals with $HO_2$ (37.2% of its total source in updrafts and 42.4% of the total source in non-updrafts) and NO reaction with $HO_2$ (32.4% of the total source in updrafts and 17.8% of its total source in the rest of the domain). In this case, the peroxy radicals are mainly formed by the oxidation of isoprene and its degradation products. The absolute value of OH reactivity is higher in thermals than in the rest of the domain.

The OH budget for the anthropogenic case (Fig. 7b) shows that the chemical reactivity is higher inside thermals at the surface
compared to the rest of the domain. The budget is largely dominated by the production of OH by $NO+HO_2$ (79.2% of its total source in updrafts and 71.2% of the total source in non-updrafts) and by $O^1D+H_2O$ (14.4% of the total source in updrafts and 18.5% of its total source in the rest of the domain). Over the whole domain, $ALD2+OH$ (21.6% of the total loss in updrafts and 26.0% of the total loss in the rest of the domain) is the most important sink at the surface and at 500 m, followed closely by the oxidation of carbon monoxide (17% of the destruction of OH in thermals and 18.8% in non-updrafts).

The chemical budget at 1200 m (Fig. 8), namely at the top of the boundary layer, allows the investigation of chemical reactions inside the ascending air parcel lifted by thermals and its comparison with its surroundings. For the biogenic case, the major OH reactants in the thermals have a chemical lifetime greater than the turbulence timescale. At this altitude, only species whose Damkhöler numbers are lower than 1 are present in sufficient amounts to react with the OH radical. For example, carbon monoxide (26.2% of total OH destruction in thermals and 36.6% in the surroundings) is the major sink, but also
methane (11.8% of total OH loss in updrafts and 18.4% in the environment). Chemical compounds with a secondary source like formaldehyde and C>2 aldehydes (ALD2) are other important sinks at 1200m. Isoprene, the major OH reactant close to the surface, is present only in thermals at this altitude due to its reaction with OH in the ascending air parcel and consumes 11.8% of OH in thermals. OH production by $NO+HO_2$ reaction is null inside updrafts and low in the non-updraft area (3.2% of

the total OH production) due to NO destruction in updrafts. The reaction between hydroxyl radicals $RO_2$, secondary products, with $HO_2$ is a major OH source in thermals (49.7% of total production) but also in the surroundings (36.1% of OH production in updraft-free region). Production of OH by $O^1D + H_2O$ or $H_2O_2$ photolysis are similar in magnitude in thermals and in the non-thermal areas. The production and destruction terms are higher in the thermals compared to these terms in non-updrafts

due to higher concentrations of OH reactants inside the thermals, but lower than at 20m.

Regarding the anthropogenic case, species whose lifetimes are higher than the turbulence timescale are major OH reactants at 1200m. Carbon monoxide contributes 22% of the OH destruction in thermals and 34.3% in the rest of the domain. As with the biogenic case, chemical compounds with a secondary source are important OH sinks like formaldehyde (11.1% in thermals and 11.0% in updraft-free regions) and ALD2 (26.4% of total OH destruction in updrafts and 11.8% in the surroundings),

corresponding to the major OH destruction term at the top of the boundary layer. The OH production terms in the anthropogenic case are similar in thermals compared to the surface with a high contribution of $NO + HO_2$ reaction (66.9% of total OH production), followed by $O^1D + H_2O$ reaction (15.3%) and $RO_2 + HO_2$ (10.6%). In the rest of the domain, $NO + HO_2$ contribution drops to 16.9% while $O^1D + H_2O$ (32.8%) and $RO_2 + HO_2$ (31.1%) are major production terms.

The differences between the OH reactivity at the surface and the top of the boundary layer are mainly driven by the changes

in chemical mixing ratios of precursors caused by chemical reactions and consequently by their Damkhöler number, and by the secondary products formed inside the thermals.

### 3.2.2 OH reactivity in the convective boundary layer

The OH reactivity for the biogenic case (Eq. 4) at 20 m is maximum around midday and is equal to 6.0 $s^{-1}$, 4.25 $s^{-1}$ and 4.55 $s^{-1}$ respectively in updrafts, non-updrafts and averaged over the domain (Fig. 9a). This feature is linked to the photochemical

maximum activity at noon, and the diurnal cycle of emissions amplifies this phenomenon. At that time and near the surface, many chemical compounds are available to react with OH, which leads to a high value of reactivity. The values in updrafts are higher than outside due to higher reactant mixing ratios inside thermals (Fig. 5a). The arithmetical difference between updrafts and non-updrafts reaches a maximum at 1200 UTC and is about 1.75 $s^{-1}$. However, as the thermals occupy less than 15% of the domain (Fig. 3), the evolution of the domain-averaged OH reactivity is very similar to that linked to updraft-free regions.

The diurnal cycle of OH total reactivity with a maximum around midday in a biogenic environment is well documented in the literature on observations of OH reactivity for a Mediterranean forest (Zannoni et al., 2016), for temperate forests (Sinha et al., 2008; Ramasamy et al., 2016) and for tropical forests (Nölscher et al., 2016; Williams et al., 2016). The values of OH reactivity in or outside thermals in the present study are the lower bounds of measurements taken over forests and gathered in Yang et al. (2016), between 1 and 76 $s^{-1}$.

The mean relative error made on OH reactivity is calculated by Equation 10 (Fig. 10). This diagnostic includes the segregation, computed relative to the boundary layer averaged values, between OH and every one of its reactants in the chemical scheme. It is negative throughout the simulation, increases during the morning and is maximal in the early afternoon with a peak at -9% at 1430 UTC. In other words, neglecting the segregation of reactive species by turbulent mixing in the boundary layer would lead to overestimating OH reactivity by 9% at most in an environment dominated by biogenic emissions.

For the anthropogenic case, total OH reactivity (Eq. 4) in thermals and in the rest of the domain does not present a clear diurnal cycle (Fig. 9b). From the 0800 UTC value of 11.6 s$^{-1}$, the domain-averaged OH reactivity fluctuates but tends to increase to 14.2 $s^{-1}$ at 1600 UTC. The evolution of the OH reactivity in thermals is similar and ranges from 14.8 $s^{-1}$ at 0800 UTC to 17.7 $s^{-1}$ at 1600 UTC. For the same period, the values in updraft-free domains vary between 11 and 13.5 $s^{-1}$. The consequence of higher OH reactant mixing ratios in the boundary layer (Fig. 4c and d), is higher OH reactivity in this case compared to the biogenic simulation.

The mean relative error of the OH reactivity (Eq. 10) is generally positive throughout the simulation (Fig. 10). It ranges between 0 to 4% from 0900 to 1600 UTC with a maximum value of 4.5% at 1330 UTC. In this case, the turbulent mixing induces a moderate increase up to 4.5% of the total OH reactivity for two reasons. The segregation effect is limited to the last 200 m of the boundary layer. Thus averaging on the whole boundary layer cancels the extreme values. Moreover, chemicals have either negative or positive segregation towards OH that may compensate or increase the positive values simulated for ALD2 and OH (Fig. 6b).

In both cases, the occurrence and development of clouds (Fig. 6, upper panel) is concomitant with linear increases of the error made on the OH reactivity while neglecting the impact of turbulence (Fig. 10). The diurnal cycle of E$_{R_{OH}}$ in each case is correlated to the development of the convective boundary layer. Firstly, a rapid change occurs during the first hours of the simulations, characterized by the occurrence of thermals and an increase in chemical emissions for the biogenic environment. Then, E$_{R_{OH}}$ is relatively stable from the end of the morning to the middle of the afternoon, with a maximum value computed around 1400 UTC, corresponding to the maximum turbulent activity in the convective boundary layer.

## 4    Discussion

The redistribution of chemical species by the boundary layer turbulence induces a different mean reaction rate between compounds when compared to a situation in which chemical species would be perfectly mixed (Krol et al. (2000), Ouwersloot et al. (2011), Kim et al. (2012), Kim et al. (2016), Li et al. (2016) and Li et al. (2017) among others). The perfectly mixed assumption used in regional and large scale atmospheric models leads to errors on the mean reaction rates between species as the turbulent mixing occurs at scales smaller than the grid length (Vinuesa and Vilà-Guerau de Arellano, 2005). This implies that the OH total reactivity has been calculated inaccurately, in turn leading to a modification in the lifetimes of the OH reactants such as ozone and carbon monoxide.

In a biogenic environment characterized by low NO$_x$ conditions, Kim et al. (2016) found negative segregation between isoprene and OH varying between -3% near the surface to -10% in the cloud layer due to increasing OH mixing ratios in thermals with altitude. That implied positive isoprene anomalies and negative ones for OH in the frontier region between the boundary layer and the free troposphere. These features are reproduced in our biogenic simulation although our segregation values are higher in the cloud layer (Fig. 6) due to sharper gradients of OH mixing ratios at the top of the cloudy layer.

Using a simple chemistry scheme of 19 reactions representing the basic reactions of O$_3$-NO$_x$-VOC-HO$_x$ system, Ouwersloot et al. (2011) found an almost constant value of -7% for the segregation between OH and isoprene in the boundary layer over the

Amazonian forest. This was the result of positive isoprene anomalies due to transport by thermals and negative OH anomalies due to consumption therein. Negative segregation ranging from -2% to -5% inside the convective boundary layer was simulated by Li et al. (2016) and Kim et al. (2016). In the biogenic case in this study, the negative segregation of a few percent in the middle of the boundary layer is reproduced. As in this case, higher segregation values were simulated with altitude in Kim et al. (2016), especially in the cloudy layer. However, segregation computed in the cloudy layer in Kim et al. (2016) was equal to -0.1, a value lower to that computed in the biogenic case of the present study. The discrepancies in this study and Kim et al. (2016) are likely due to the vertical OH profiles. In the study by Kim et al. (2016), OH concentrations increased linearly with altitude. This implies lower OH covariances for ascending air parcels and thus lower segregation values (Eq. 2). On the contrary, in the present study, OH is relatively homogeneous in the boundary layer and a strong gradient is present only at the top of the boundary layer. This induces high covariances for OH concentrations inside air transported by thermals at the top of the boundary layer, implying higher segregation values. As segregation computed by Li et al. (2016) is not available above 1000m height, a direct comparison with results regarding the cloudy layer is not possible with the results from the biogenic case. The positive values of segregation simulated in the afternoon (Fig. 6a) in the biogenic case are not reproduced in other studies and might be the result of efficient OH recycling in ReLACS 3.0, initiated in particular inside the thermals due to peroxy radicals formed by isoprene oxidation. Indeed, this recycling is either absent or indirect in previous works like in the mechanism used by Kim et al. (2016) that produces only $HO_2$ from peroxy radicals, which may explain the discrepancies in OH covariances. Furthermore, Ouwersloot et al. (2011) and Kim et al. (2016) investigated the sensitivity of segregation to $NO_x$. It was found that different $NO_x$ levels imply differences in the segregation of OH and other compounds as they contribute to OH production.

Kim et al. (2012) used LES to study the OH budget in a biogenic environment averaged over the domain and over time from 1330 to 1430 LT with a chemical scheme adapted from MOZART v2.2. For a low $NO_x$ case with ozone mixing ratios close to 64 ppbv, they found that OH production was mostly influenced by four predominant reactions (including $O^1D$ + $H_2O$, $HO_2$ + $O_3$, $H_2O_2$ photolysis and $HO_2$ + NO). These production terms are present in the biogenic case of the present study. However, Kim et al. (2012) did not take into account recycling by peroxy radicals as the latter formed only $HO_2$ in the chemical scheme they used. Moreover, they found that OH loss by reaction with isoprene was dominant near the surface, followed by reactions with CO and formaldehyde, as in the present study. The contribution of BVOC species to OH reactivity was simulated by Li et al. (2016) for three distinct biogenic cases of the DISCOVER-AQ (Deriving Information on Surface Conditions from Column and Vertically Resolved Observations Relevant to Air Quality) campaign. The production terms of OH were not available but it was found that isoprene was a dominant sink for OH; about 25-30% of the OH reactivity was linked to BVOC reactions at 0.3 km during midday. This contribution is comparable to the percentage calculated for isoprene in the OH destruction in thermals (41.7%) or in an updraft-free area (29.3%). The contribution of formaldehyde was comparable to isoprene at that height, with HCHO mixing ratios ranging from 2 to 4.5 ppbv at the surface. The higher formaldehyde contribution of Li et al. (2016) is the result of higher concentrations than in the current work.

Yang et al. (2016) found that in forest areas the OH budget was largely dominated by isoprene and its oxidation products. For example, Zannoni et al. (2016) measured OH reactivity and the concentration of biogenic compounds over a Mediterranean

forest. They found that isoprene was the dominant sink for OH and contributed up to 74% of total OH reactivity during daytime due to its high reactivity towards OH and its high concentration over the forested area. Isoprene predominance in OH loss was reproduced in the OH budget of the biogenic environment of the present study, and for carbon monoxide. The formaldehyde mixing ratios were close to 2 ppbv, which explains that this is not a major sink like isoprene, but remains important for OH

loss through the reaction OH+HCHO. OH production in the biogenic case was similar to Kim et al. (2012), as almost the same production terms are present. However, overall production was dominated by the reaction of peroxy radicals with $HO_2$ and the ozone mixing ratios of 18 ppbv decrease the importance of $O^1D + H_2O$ source for OH.

However, the mechanism used to represent atmospheric chemistry in our simulation has fast OH cycling due to the reaction of peroxy radicals $RO_2$ with $HO_2$. This reaction yield in OH is greater than in the laboratory studies (Hasson et al., 2004;

Jenkin et al., 2007; Groß et al., 2014; Winiberg et al., 2016), implying an overestimation of OH mixing ratios, especially in the biogenic case where $RO_2$ are high. As the hydroxyl radical is recycled through the isoprene oxidation products, it tends to reduce the impact of isoprene chemistry on OH mixing ratios and thus influences the low segregation simulated in the core of the boundary layer. This is similar to a case considered by Stone et al. (2011) where an ISOPO2 + $HO_2 \rightarrow$ ISOPOOH + 3 OH recycling mechanism was proposed to increase the simulated OH concentrations and provided the best agreement

with OH observations. This reaction implied that isoprene has no net impact on OH concentrations. This conclusion was also reached by Kubistin et al. (2010) who found better agreement between simulated and measured $HO_x$ concentrations by ignoring isoprene chemistry. As the chemistry of isoprene is not well understood, the results obtained in our work are subject to these uncertainties.

Aldehydes have not been considered in previous works on segregation. Auger and Legras (2007) calculated segregation at

250 m and 1100 UTC between each species of the chemical model CHIMERE used in their simulation. They found segregation ranging from 0 to -1% between OH and acetaldehyde. This is comparable to range of results of the anthropogenic case of the present study (Fig. 6b) considering the incomplete mixing between OH and C>2 aldehydes at the same height.

In the anthropogenic simulation, the high values of segregation computed at the top of the boundary layer are coincident with the presence of clouds. In the absence of aqueous-phase chemistry, clouds impacts on chemical species are dynamical and

photochemical. They modify heat and moisture fluxes at their surroundings (Vilà-Guerau de Arellano et al., 2005) and thus the transport of compounds, as noted by Kim et al. (2012) who demonstrated that clouds presence could increase transport of chemicals to 1000m. In this set of simulations, the chemical impact of clouds on species involves photolysis rates as they are corrected at every time step due to the presence of clouds according to the work of Chang et al. (1987). At each point of the domain, photolysis rates are increased above clouds and decreased below them.

Another effect of clouds on the atmospheric chemistry lies in isoprene emissions, as demonstrated by Kim et al. (2012). As isoprene emissions are dependent on incoming radiation and temperature near the surface, cloud shading could decrease the amount of isoprene emitted. Kim et al. (2012) showed that isoprene concentrations are decreased by 10% and OH concentrations increased by 5% in the boundary layer when isoprene emissions are reduced by up to 10%. The ultimate impact on segregation cannot be anticipated since it corresponds to two compensating effects.

Finally, clouds impact atmospheric chemistry through aqueous phase reactivity. Aqueous-phase chemistry was not considered here, nor were the exchanges between gas and aqueous phases. However, it could have an impact on soluble species mixing ratios, such as formaldehyde and $H_2O_2$ through the capture and degassing cycles of these compounds. Lelieveld and Crutzen (1990) showed a decrease in oxidative capacity of the atmosphere through aqueous-phase reactions via a significant decrease

in ozone mixing ratios, but also OH, formaldehyde and nitrogen oxides. However, the effects of aqueous-phase chemistry on gas-phase compound concentrations are various (Barth et al., 2003) and OH concentrations could decrease in clouds (Mauldin et al., 1997). This result was confirmed by the study of Commane et al. (2010) who found that $HO_x$ concentrations decreased in clouds. Recently, Li et al. (2017) studied segregation effects in a biogenic environment when aqueous-phase chemistry is included. They found that isoprene concentrations are increased by up to 100% while OH concentrations decreased by 18%,

resulting in a maximum segregation of 55% in the cloudy layer. In the anthropogenic environment, segregation effects are expected to be enhanced due to the decrease of OH concentrations in gaseous phase in the cloud layer, reducing the cleansing capacity of the atmosphere.

Several instrumental studies examined OH total reactivity in urban environments, and more especially the OH budget. Hansen et al. (2015) found that $NO_x$ contribution (50 to 55%) in an urban environment had the greatest effect on OH loss,

considering $NO_x$ mixing ratios between 10 and 300 ppbv. The next most important contribution came from OVOCs (especially aldehydes and ketones) leading to 15 to 25% of OH destruction with mixing ratios close to 20 ppbv. Some discrepancies exist between this experimental study and the OH budget for the anthropogenic case in our study (Fig. 7b). Given the $NO_x$ mixing ratios simulated in the present study (< 3 ppbv at midday), they are not an important sink for OH as measured by Hansen et al. (2015). However, the range of OVOC contribution reported by Hansen et al. (2015) is comparable to what was found for ALD2

in the anthropogenic case.

In the city of Tokyo, Sadanaga et al. (2004) found that the OH budget was dominated by NMHCs and by OVOCs. In their study, the OVOC group including acetaldehyde, formaldehyde, methanol, ethanol and acetone, contributed up to 18% of OH reactivity. If aggregated, the species constituting the NMHC group are the predominant sink for OH in our anthropogenic simulation, as observed by Sadanaga et al. (2004). However, C>2 aldehydes contribute more to OH destruction in our study

than in that of Sadanaga et al. (2004). This could result from the cyclic boundary conditions prescribed at the borders of our domain, which can cause the ageing of air masses and increase the mixing ratios of secondary products such as aldehydes.

Lelieveld et al. (2016) studied the global distribution and budget of OH radical using the EMAC model (ECHAM/Messy Atmospheric Chemistry) coupled with the Mainz Organics Chemistry (MOM). They found that the annual mean OH reactivity near the surface ranged from 10 to 20 $s^{-1}$ in southern West Africa, which is in agreement with the results of the anthropogenic

case but slightly higher than the values obtained in our biogenic case. However, Nölscher et al. (2016) studied, by means of observations, the effects of seasonality in rainforest air reactivity and noticed that total OH reactivity was much lower during the wet season than during the dry season due to lower temperatures and incoming radiation. More specifically, measurements of reactivity performed during the wet season varied between 6 and 12 $s^{-1}$ with an average value of 9.9±5.2 $s^{-1}$ at 24 meters, which is much closer to the Meso-NH model results. Moreover, the error made on the total OH reactivity neglecting the

turbulent mixing could cumulate with the uncertainties reported in the literature regarding OH reactivity techniques such LIF

with a flow tube (from 10 to 15%, Kovacs and Brune (2001)), LP-LIF (from 10 to 20%, Sadanaga (2004)) and the CRM measurement method (15 to 20%, Sinha et al. (2008)). In addition, uncertainties on reaction rate constants are also present in chemical schemes, including those used by numerical models. These uncertainties on reaction rate coefficients range from 5 to 15%, as suggested by Atkinson et al. (2006). It is likely that the unaccounted fraction of OH reactivity reported in the literature may be explained at least partially by a combination of the following phenomena similar in intensity: turbulence effects on chemical reactivity and uncertainties on the OH reactivity measurements and reaction rate coefficients.

## 5 Conclusions

A numerical simulation coupled with a realistic chemical mechanism was performed with the atmospheric model Meso-NH to study the impact of thermals on the oxidizing capacity of the atmosphere. The fine grid resolution of the LES version of the model made it possible to explicitly resolve the thermals spatially and temporally. Identification of thermals was based on a conditional sampling method relying on a first-order decay passive scalar. The impact of turbulent mixing on chemical species redistribution, and of the consequences on OH reactivity was determined in a natural environment and a more contrasted urban case.

The differentiated transport by thermals is dependent on the chemical lifetime of the compounds, represented by the Damköhler number. This transport induces inhomogeneous mixing of the species within the boundary layer, with an impact on the mean chemical rate between reactive species. In both natural and urban environments, the top of the boundary layer was the region with the highest calculated segregation intensities but of the opposite sign. Between isoprene and the OH radical, an effective maximum decrease of 30% of the reaction rate was calculated at the top of the boundary layer in a biogenic environment compared to a perfectly mixed case. In the urban case, the reduction of the mean chemical reaction rate between the OH radical and C>2 aldehydes reached 8% at the surface in the early morning while this reaction increased to 16% at the top of the boundary layer during most of the simulation. Thermals transporting species emitted at the surface can lead to different chemical regimes inside updrafts and the environment. For both cases, the surface and the thermals are the preferential reaction zones, with highest chemical reactivity. This was especially the case for the OH radical whose precursors are either transported by thermals or created inside them. OH reactivity was always higher by 15 to 40% inside thermals compared to their surroundings depending on the time of day. For the natural case, the major OH precursors close to the surface were radicals originating from the oxidation of isoprene and its degradation products whereas $O_1D+H_2O$ reaction became more predominant with increasing altitudes. In the urban case, OH was mainly produced through the reaction between $HO_2$ and NO, at the surface or higher in the boundary layer. This led to a higher oxidation capacity in the air transported by thermals for both cases.

The overall impact of turbulence on OH concentrations and reactivity at the domain scale differs depending on the chemical environment considered. In a biogenic environment with low OH mixing ratios varying from 0.18 to 0.24 ppt, turbulent structures had little impact on the redistribution of OH in the boundary layer. This was due to an efficient OH recycling initiated by peroxy radicals formed by BVOC oxidation. In the anthropogenic case, OH mixing ratios ranged from 0.26 to 0.50 ppt. The

turbulence significantly impacted the spatial distribution of OH and its precursors in the boundary layer, with higher mixing ratios in thermals.

The mean relative error on domain-averaged OH reactivity revealed that effective OH reactivity (taking into account segregation by turbulent motions) in the biogenic case was up to 9% below the OH reactivity calculated based on averaged boundary layer mixing ratios. Accounting for inhomogeneous mixing between OH and its reactants (primarily isoprene) in a regional or global model could lower the calculated OH reactivity and increase the discrepancies with observed OH reactivities. In the urban environment, the mean relative error was slightly positive which means that air mass segregation by turbulence increases OH reactivity. Considering the effect of turbulent motions could reduce the gap between modelled and observed OH reactivity. However, segregation alone is unlikely to resolve the underestimation between observed and modeled OH reactivity.

This study addressed the impact of moist thermals on the oxidative capacity of the atmosphere on two contrasted chemical situations in a wet environment represented by the monsoon flow. However, Nölscher et al. (2016) observed a substantial seasonal cycle in OH reactivity over an Amazonian forest ranging from 10 s s$^{-1}$ in the wet season to 62 s s$^{-1}$ in the dry season. It would be interesting to assess the impact of turbulent mixing on chemistry in the dry season over the southern part of West Africa. Moreover, the influence of urban area in this study was only linked to chemical emissions. This work should be repeated by taking into account the dynamical forcing due to the presence of the urban area as in Ouwersloot et al. (2011) in which the authors introduced heterogeneous surface conditions over forest and savannah patches. They showed that the difference in buoyancy fluxes at the surface could have an impact on the redistribution of species, and thus on segregation. Finally, the presence of clouds was considered only from a dynamical view. Adding aqueous phase chemistry in these simulations could provide further insight into the impact of moist thermals on chemical reactivity.

*Acknowledgements.* The authors are very grateful to Mat Evans for his helpful comments and discussions. The research leading to these results has received funding from the European Union 7th Framework Programme (FP7/2007-2013) under Grant Agreement no.603502 (EU project DACCIWA: Dynamics-aerosol-chemistry-cloud interactions in West Africa). This work was performed using HPC resources from GENCI-CINES (Grant 2016-A0010100005) and CALMIP Toulouse (P12171, P17015). The authors acknowledge ECCAD (http://eccad.aeris-data.fr/) for the archiving and distribution of emissions data.

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

**Table 1.** Initial vertical profiles of mixing ratios and associated profiles. Numbers in brackets refer to (1) uniform profile, (2) stratospheric profile (initial profile multiplied by 1 from 0 to 2000m, then by 0.5 from 3000m to 13000m, by 0.75 at 14000m and 1 above), (3) boundary-layer profile (multiplied by 1 from 0 to 1000m, by 0.10 from 2000 to 13000m and 0.05 above). Chemical names are those used in the ReLACS3 chemical mechanism. ALD2 corresponds to aldehydes C>2, ALKL to lumped alkanes $C_2C_6$, ALKH to lumped alkanes $C_7C_{12}$, OLEL to lumped alkenes $C_3$-$C_6$, OLEH to lumped alkenes C>6, ETHE to ethene, ISOP to isoprene, AROL to lumped low SOA yield aromatic species, AROH to lumped high SOA yield aromatic species , AROO to lumped phenolic species, ARAC+ to lumped aromatic monoacids, MEOH to methanol, ARAL to lumped aromatic monoaldehydes, MVK to methyl-vinyl-ketone, MCR to methacrolein, HCHO to formaldehyde, KETL to lumped ketones $C_3$-$C_6$ and PAN2 to peroxy acetyl nitrate

| Species | Initial mixing ratio | Species | Initial mixing ratio |
|---------|---------------------|---------|---------------------|
| $O_3$ | 21.19 ppb (2) | $HO_2$ | 2.48 ppt (1) |
| OH | 0.07 ppt (1) | NO | 55.25 ppt (2) |
| CO | 149.23 ppb (3) | HCHO | 747.47 ppt (3) |
| ALD2 | 896.82 ppt (3) | PAN2 | 35.70 ppt (3) |
| ALKL | 282.90 ppt (3) | ALKM | 3.67 ppt (3) |
| ALKH | 0.60 ppt (3) | ETHE | 277.12 ppt (3) |
| OLEL | 104.71 ppt (3) | OLEH | 0.94 ppt (3) |
| ISOP | 1.23 ppb (3) | AROH | 53.29 ppt (3) |
| AROL | 14.11 ppt (3) | AROO | 4.66 ppt (3) |
| ARAC+ | 0.69 ppt (3) | ARAL | 1.98 ppt (3) |
| MEOH | 564.54 ppt (3) | KETL | 72.69 ppt (3) |
| MVK | 537.74 ppt (3) | MCR | 268.87 ppt (3) |

**Table 2.** Emission values for the biogenic and the anthropogenic cases, in kg.m$^2$.s$^{-1}$. For compounds in bold characters, Gaussian shape emissions were set and only the maximum value is indicated here. Please see text for details.

| Chemical species | Molar mass $(g.mol^{-1})$ | Biogenic patch emissions $(kg.m^{-2}.s^{-1})$ | Anthropogenic patch emissions $(kg.m^{-2}.s^{-1})$ |
|---|---|---|---|
| NO | 30 | $2.23 * 10^{-11}$ | $5.68 * 10^{-10}$ |
| NO$_2$ | 46 | - | $2.44 * 10^{-10}$ |
| CO | 28 | $2.13 * 10^{-11}$ | $2.43 * 10^{-08}$ |
| ETHE | 28 | $7.72 * 10^{-12}$ | $8.06 * 10^{-10}$ |
| OLEL | 70 | $3.26 * 10^{-12}$ | $8.68 * 10^{-10}$ |
| OLEH | 126 | $4.20 * 10^{-14}$ | $1.91 * 10^{-11}$ |
| ALKL | 72 | $2.72 * 10^{-14}$ | $3.92 * 10^{-10}$ |
| ALKM | 128 | $6.13 * 10^{-15}$ | $1.31 * 10^{-10}$ |
| ALKH | 226 | $8.76 * 10^{-16}$ | $2.16 * 10^{-11}$ |
| AROH | 134 | $2.75 * 10^{-13}$ | $2.95 * 10^{-10}$ |
| AROL | 120 | - | $1.98 * 10^{-10}$ |
| AROO | 122 | - | $8.94 * 10^{-11}$ |
| ARAC+ | 136 | - | $1.32 * 10^{-11}$ |
| ARAL | 120 | - | $3.81 * 10^{-11}$ |
| ALD2 | 86 | $2.99 * 10^{-12}$ | $2.41 * 10^{-10}$ |
| HCHO | 30 | $8.32 * 10^{-13}$ | $8.39 * 10^{-11}$ |
| ACID | 74 | - | $1.12 * 10^{-10}$ |
| ORA1 | 46 | $6.24 * 10^{-13}$ | $5.75 * 10^{-10}$ |
| ORA2 | 60 | $6.24 * 10^{-13}$ | $6.01 * 10^{-10}$ |
| KETL | 86 | $6.80 * 10^{-13}$ | $8.04 * 10^{-12}$ |
| KETH | 114 | $2.51 * 10^{-14}$ | $4.14 * 10^{-13}$ |
| MEOH | 32 | $3.61 * 10^{-11}$ | $2.64 * 10^{-11}$ |
| ETOH | 46 | $2.15 * 10^{-12}$ | $2.95 * 10^{-10}$ |
| ALCH | 102 | - | $1.45 * 10^{-10}$ |
| ISOP | 68 | $2.76 * 10^{-10}$ | - |
| BIOL | 154 | $3.29 * 10^{-11}$ | - |
| BIOH | 88 | $4.94 * 10^{-11}$ | - |
| SO$_2$ | 64 | - | $1.60 * 10^{-10}$ |
| NH$_3$ | 17 | - | $6.02 * 10^{-11}$ |
| MTBE | 88 | - | $2.30 * 10^{-10}$ |

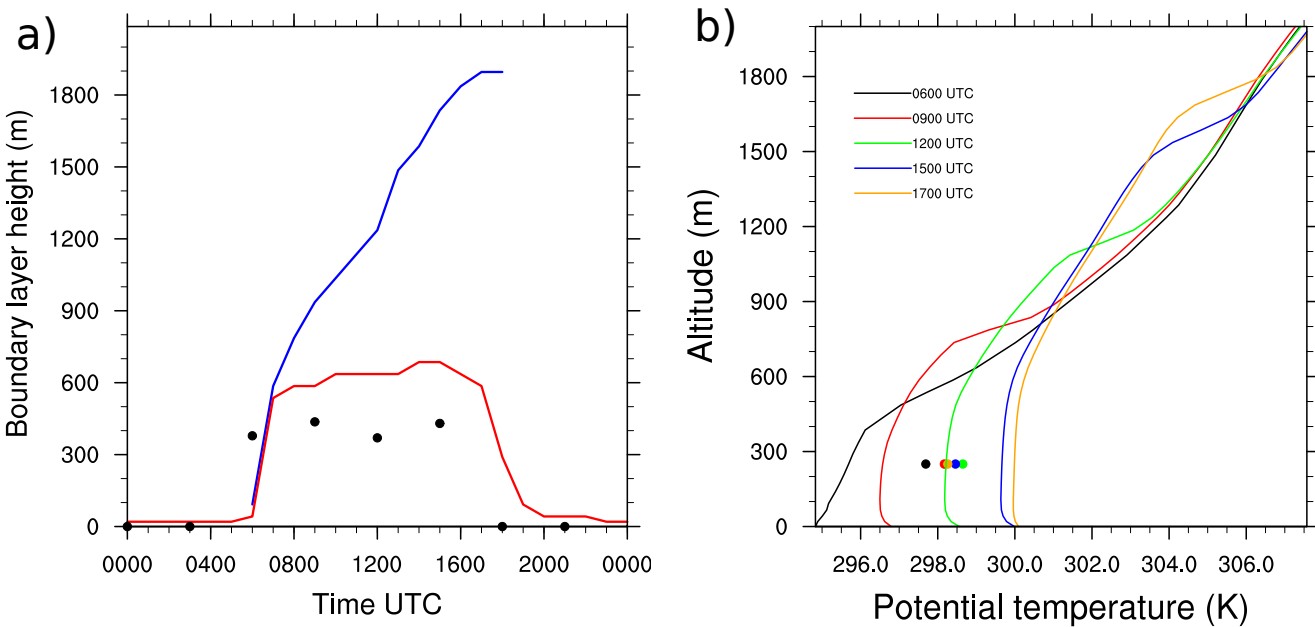

**Figure 1.** (a) Boundary layer height diurnal evolution computed from the bulk-Richardson method (red), the tracer method (blue) and composite values of boundary layer height for the sOP-2 period (08/01/06-08/15/06) based on Gounou et al. (2012) (black dots). (b) Potential vertical temperature profiles at 0600 (black), 1000 (red), 1400 (green) and 1700 UTC (blue)

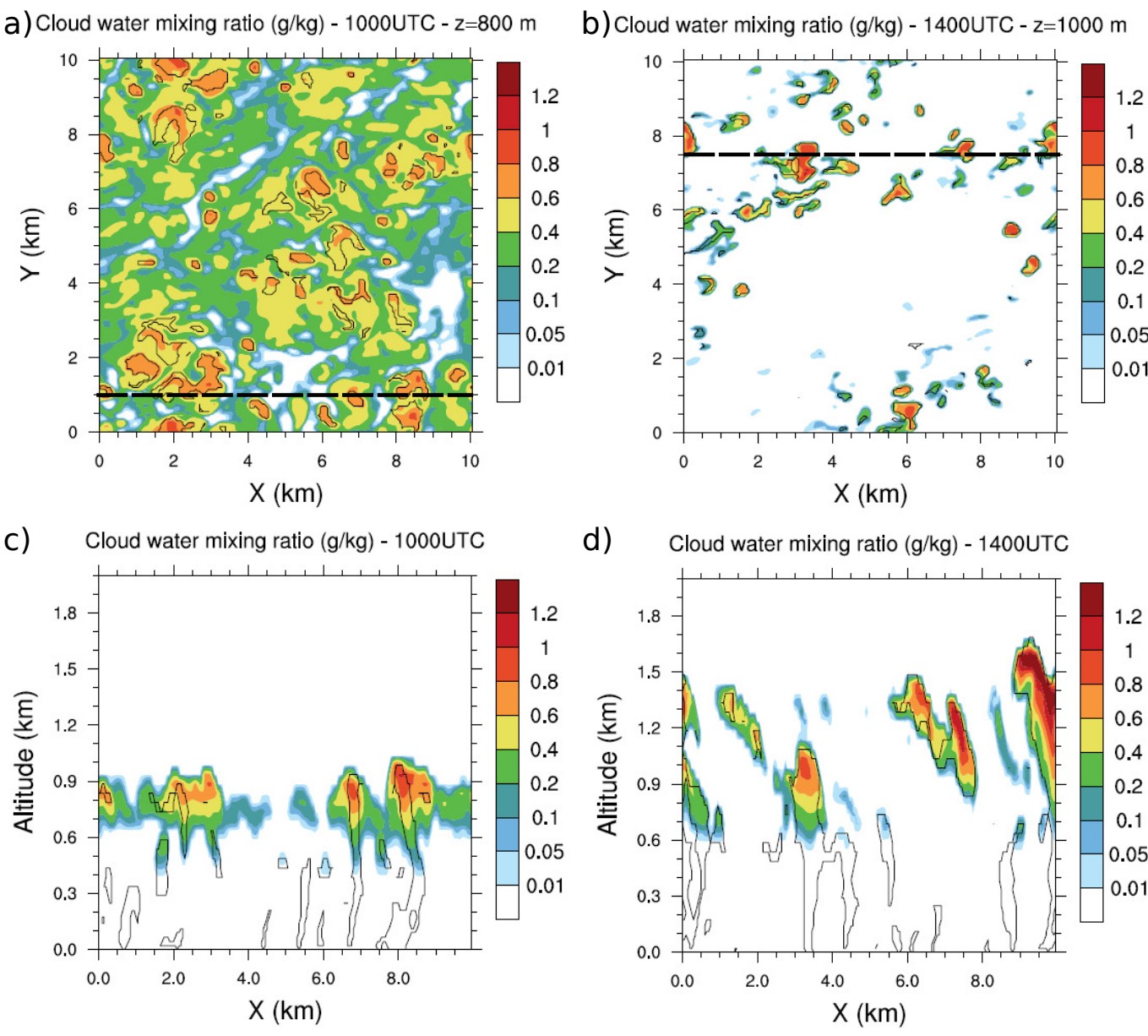

**Figure 2.** (a) Cloud water mixing ratio horizontal cross-sections at 1000 UTC and 800m height and (b) at 1400 UTC and 1000m height . Black dashed lines represent the vertical cross-sections at (c) 1000 UTC and y=1km and at (d) 1400 UTC and y=7.5km. Black isolines denote thermals identified by the conditional sampling method.

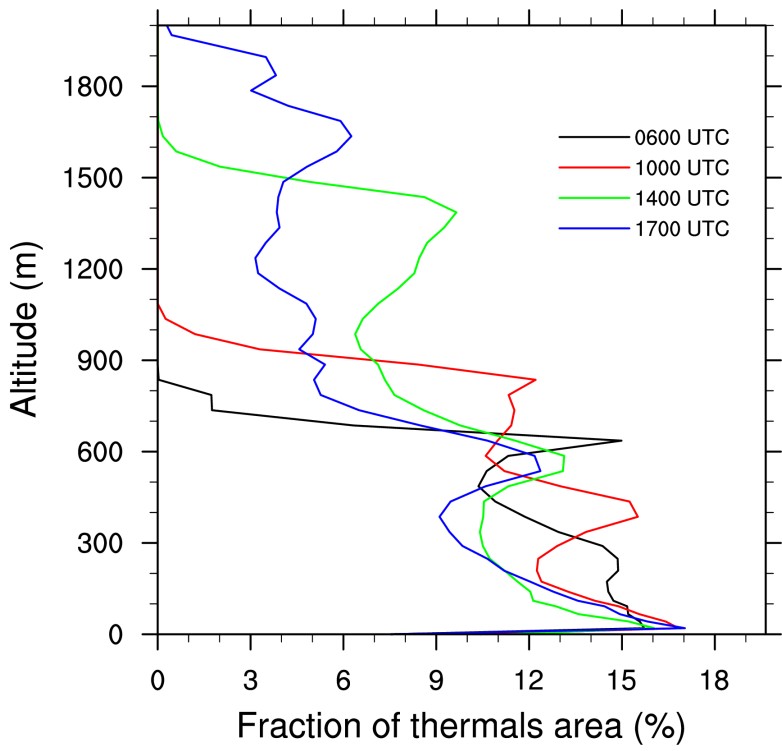

**Figure 3.** Vertical profiles of the fraction occupied by thermals at 0600 (black), 1000 (red), 1400 (green) and 1700 UTC (blue).

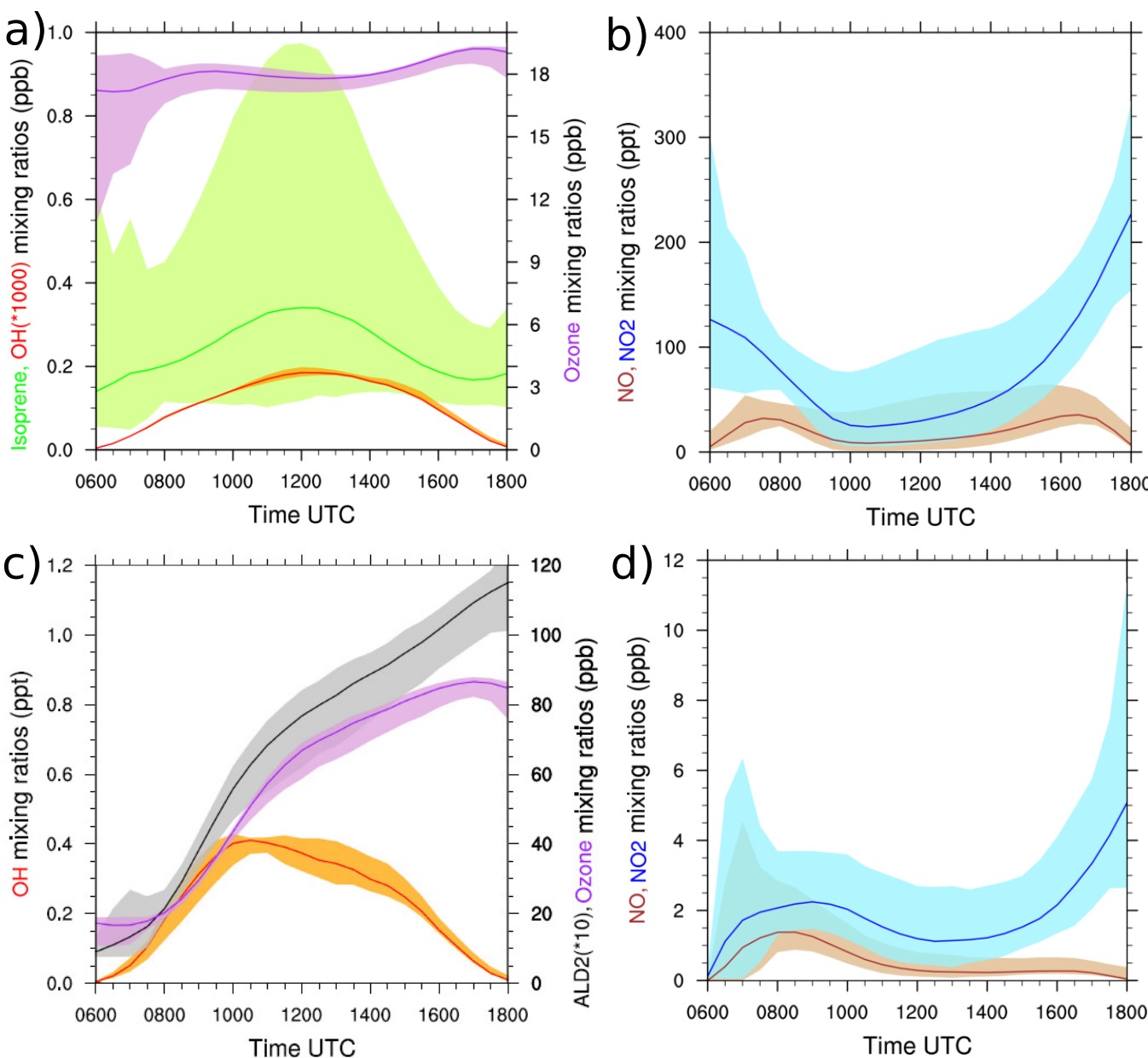

**Figure 4.** (a)-(b) LES simulated diurnal cycles for isoprene (green), OH (red), ozone (purple), NO (brown), NO$_2$ (blue), (c)-(d) ALD2 (grey), OH (red), ozone (purple), NO (brown) and NO$_2$ (blue). (a)-(b) corresponds to the biogenic environment, (c)-(d) to the anthropogenic environment. The shaded areas denote the minimum and maximum of mean vertical profiles between the surface and 600m height, and the colored lines correspond to the averages of these profiles

**Table 3.** Measurements collected during the AMMA campaign over forested areas (upper part) and over cities (lower part). The last column corresponds to the concerned period.

| Species | Mixing ratio | Altitude | Location | References | Comments |
|---|---|---|---|---|---|
| $O_3$ | 24 ppbv | 500 m | $10^o$N | Commane et al. (2010) | 1340 UTC - 17/08 |
| | 25 ppbv | <900 hPa | $10^o$N | Reeves et al. (2010) | Median value (20/07 - 21/08) |
| | 25-30 ppbv | 300 - 1700 m | 12-13$^o$N | Borbon et al. (2012) | 0800 -1800 UTC |
| | 22 ppbv | <700 m | 7.2-13.1$^o$N | Murphy et al. (2010) | Mean value (17/07 - 17/08) |
| Isoprene | 1.2 ppbv | <700 m | $10^o$N | Ferreira et al. (2010) | 1345 UTC - 17/08 |
| | 0.604 ppbv | <700 m | 7.2-13.1$^o$N | Murphy et al. (2010) | Mean value (17/07 - 17/08) |
| | 1 - 1.5 ppbv | 400 -1450 m | $10^o$N | | 1200 UTC - 17/08 |
| | 0.2-1.5 ppbv | Surface | 9.42$^o$N;1,44$^o$E | Saxton et al. (2007) | Composite diurnal cycle (07/06 - 13/06) |
| OH | 0.05 - 0.15 pptv | 500 m | $10^o$N | Commane et al. (2010) | 1345 UTC - 17/08 |
| $NO_x$ | 0.2 ppbv | <900 hPa | $10^o$N | Reeves et al. (2010) | Median value (20/07 - 21/08) |
| | 0.1 ppbv | <700 m | 9$^o$N | Delon et al. (2010) | Mean value (05/08 - 17/08) |
| $O_3$ | 40 ppbv | <2km | Cotonou | Ancellet et al. (2009) | 08/19 p.m. flight |
| | 24-30 ppbv | 0-2 km | Cotonou | Thouret et al. (2009) | Mean value (August 2006) |
| | 26 ppbv | <700m | Lagos | Murphy et al. (2010) | B229 flight - Mean value (17/07 - 17/08) |
| | 31 ppbv | | Niamey | | |
| $NO_x$ | 1 ppbv | <2km | Cotonou | Ancellet et al. (2009) | 08/19 p.m. flight |
| OH | 0-0.5 pptv | 500m | Lagos | Commane et al. (2010) | B229 flight |

**Table 4.** Simulated Damkhöler number of chemical species averaged from the surface to 0.6 km at 1200 UTC for the two cases calculated with $\tau_{turb} \approx 20$ min.

| | Damkhöler number (unitless) | |
|---|---|---|
| | Biogenic case | Anthropogenic case |
| OH | 4402.7 | 12002.5 |
| $HO_2$ | 50.4 | 137.5 |
| ISOP | 0.54 | 1.08 |
| $O_3$ | 0.66 | 0.79 |
| NO | 80.2 | 81.0 |
| $NO_2$ | 24.3 | 17.9 |
| ALD2 | 0.086 | 0.17 |

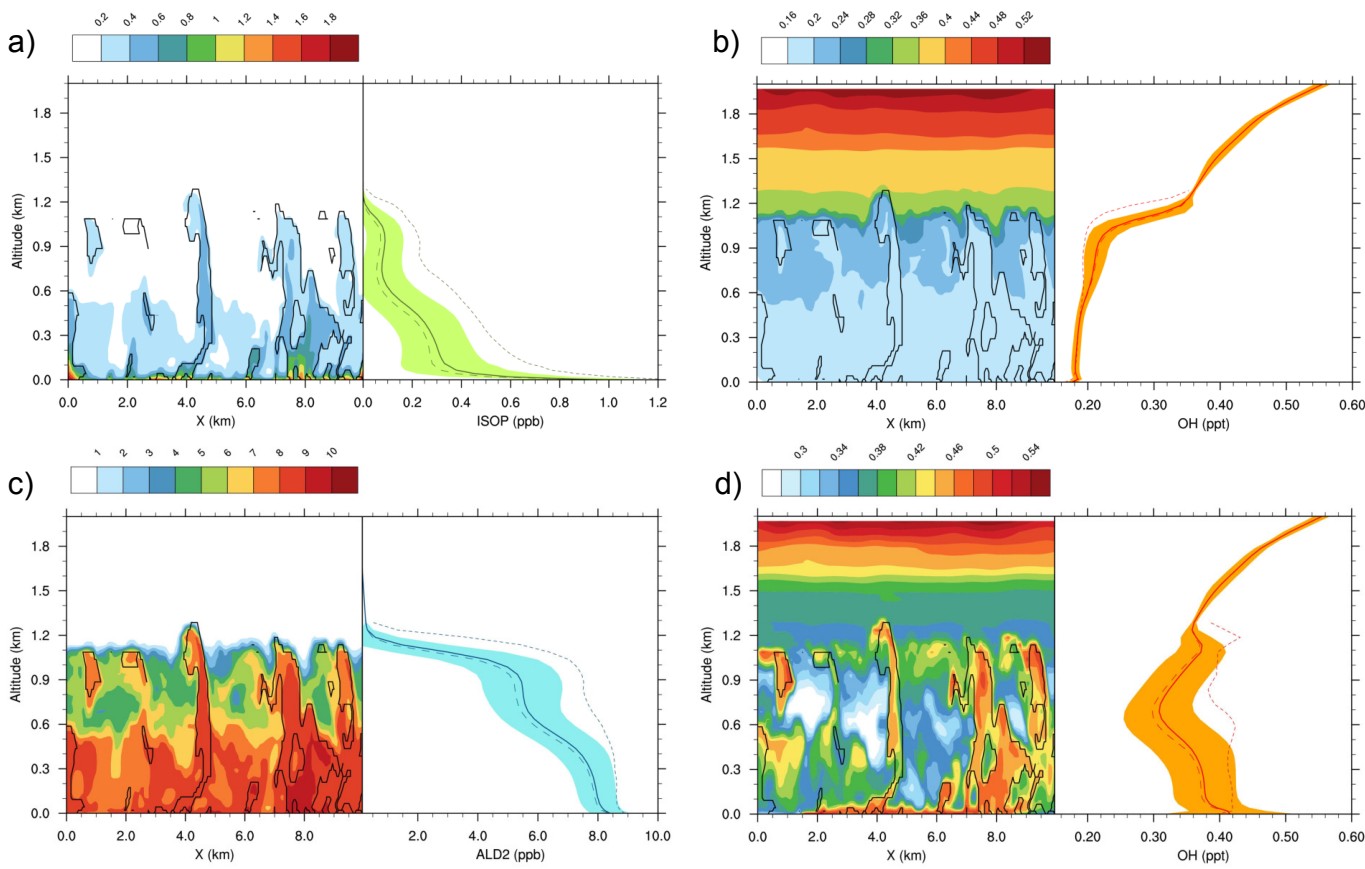

**Figure 5.** (a) Isoprene (ppb) and (b) OH radical (ppt) mixing ratios for the biogenic case at 1200 UTC. (c) ALD2 (ppb) and (d) OH radical (ppt) mixing ratios for the anthropogenic case at the same time. For each figure, the left part consists of a vertical cross-section at the middle of the domain (y = 5km). Black isolines denote thermals identified by the conditional sampling method. In the right part of each panel, the shaded areas denote twice the standard deviation at a given altitude and the line represents the horizontal average over the domain. The dashed line corresponds to the average in the environment and the dotted lines inside thermals.

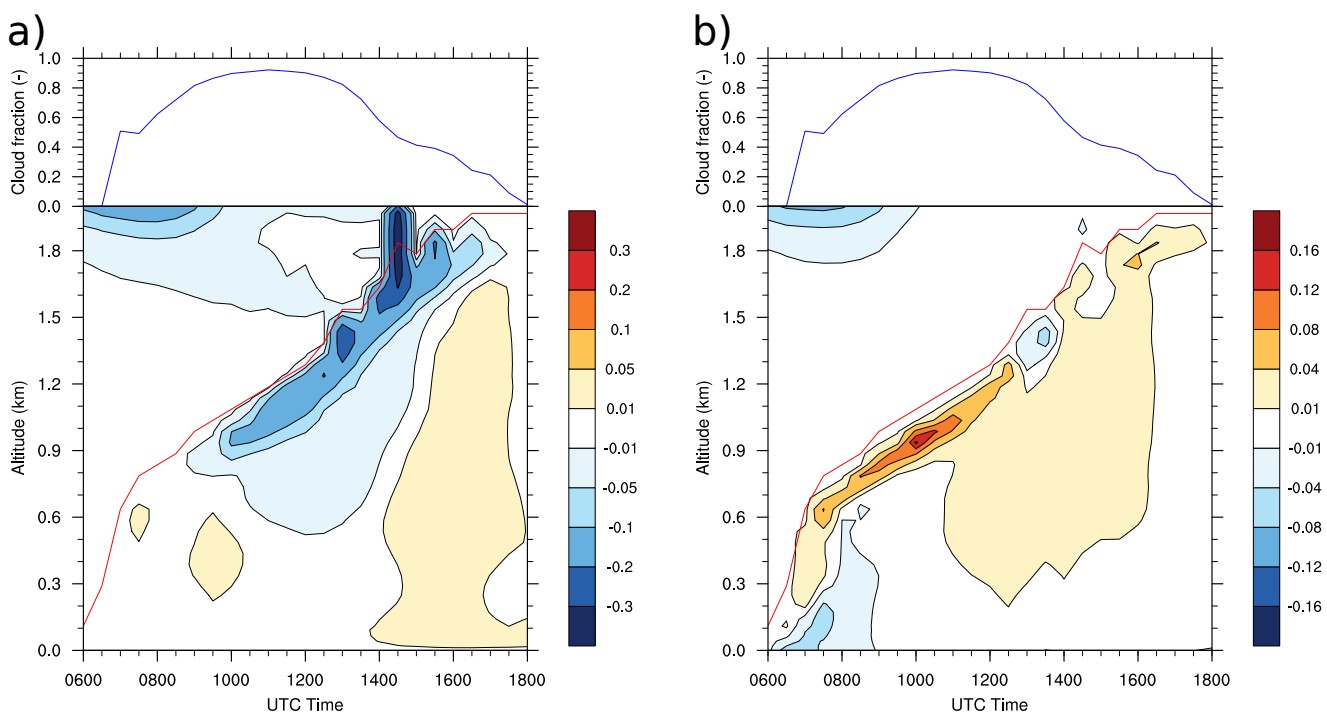

**Figure 6.** (a) Diurnal evolution of the segregation coefficient for isoprene and OH in the biogenic case and (b) for ALD2 and OH in the anthropogenic case. The red line represents the boundary layer height as determined by the tracer approach.

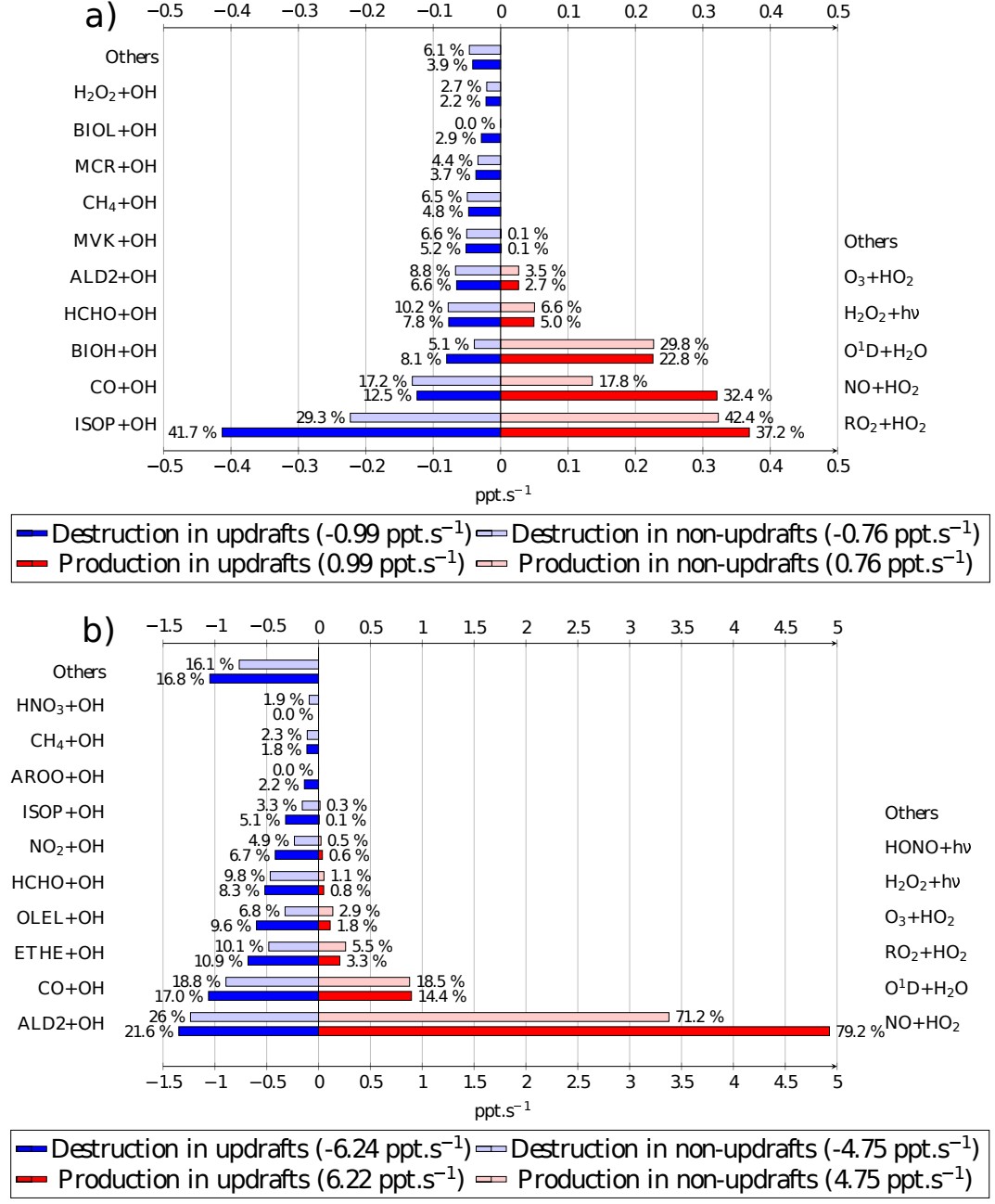

**Figure 7.** OH radical chemical budget at 1200 UTC averaged at 20 m for the biogenic (a) and the anthropogenic case (b). Source terms are in red for updrafts and pale red for non-updrafts. Destruction terms are in blue for updrafts and pale blue for non-updrafts. The bar lengths determine the absolute values and the relative contributions for destruction and production are given by the percentage near each bar. Numbers between parentheses in the legend box are the OH production and destruction rates in updrafts and in non-updrafts. Chemical names correspond to the names given in the ReLACS 3 mechanism.

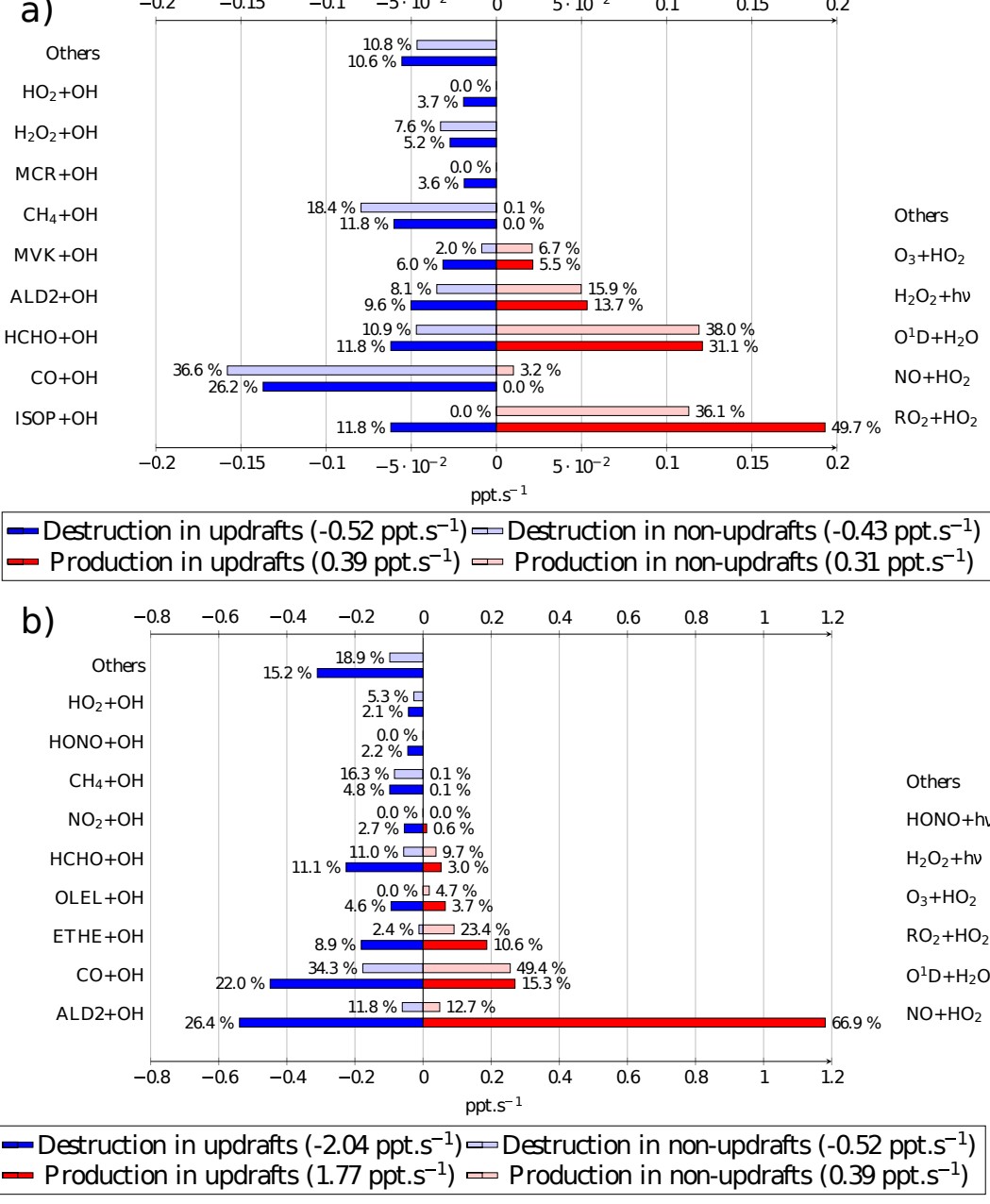

**Figure 8.** Same as figure 7 for the height of 1200m.

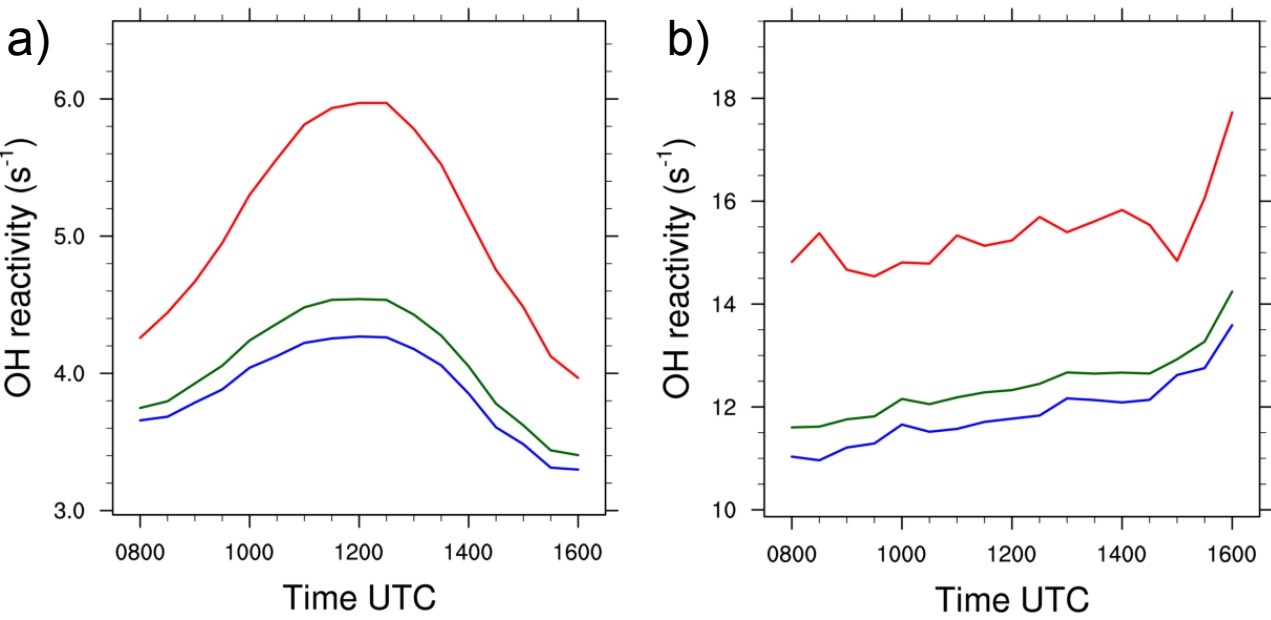

**Figure 9.** (a) OH reactivity (s$^{-1}$) in thermals (red), domain averaged (green) and in non-thermals region (blue) averaged at z = 20 m for the biogenic and (b) the anthropogenic case.

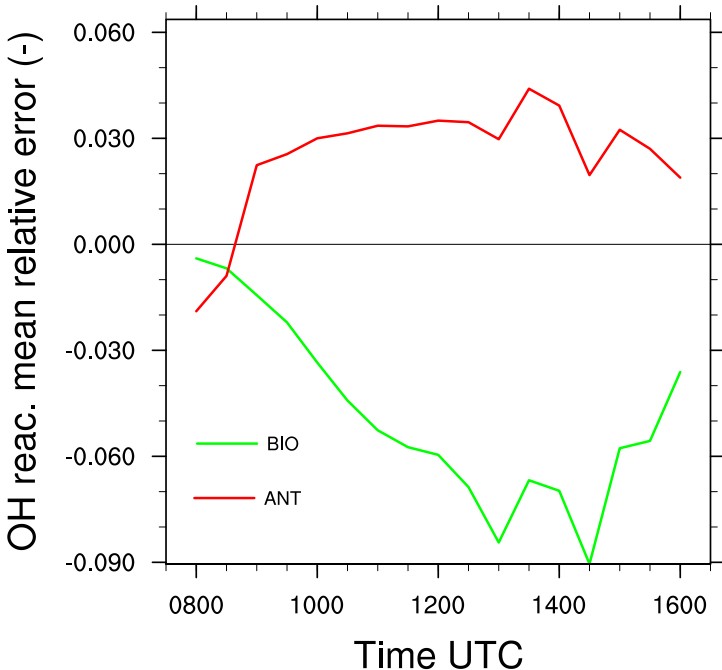

**Figure 10.** Mean relative error $E_{R_{OH}}$ made on the OH reactivity by neglecting turbulent motions for the biogenic case (green) and the anthropogenic case (red).