# Peer review of "LES study of the impact of moist thermals on the oxidative capacity of the atmosphere in southern West Africa"

_Atmospheric Chemistry and Physics, 2017_

## Referee Comment (RC1) · Anonymous Referee #1 · 3 Jan 2018

Review of 'LES study of the impact of moist thermals on the oxidative capacity of the atmosphere in southern West Africa' by Fabien Bosse et al.

Atmos. Chem. Phys. Discuss., https://doi.org/10.5194/acp-2017-969

The authors present a study of the impact of turbulent mixing and segregation of OH radicals and OH sinks on modelled reaction rates and OH reactivity. A number of simulations are presented for various cases, including regions dominated by biogenic and anthropogenic emissions. However, the paper is lacking an overall summary and conclusion on the impact of these effects on model simulations for [OH] and OH reactivity.

The introduction provides an overview of some previous OH reactivity measurements

and studies investigating the impacts of air mass segregation. However, the studies investigating air mass segregation are typically concerned more with simulations of OH radical concentrations rather than OH reactivity. Are measurements of OH reactivity typically made on a timescale sufficiently rapid for the impacts of turbulence and segregation to be investigated?

The discussion and conclusion would benefit from a discussion of the overall impacts of neglecting effects of turbulence and air mass segregation. Are the differences in [OH] and OH reactivity significant? How do they compare to measurement uncertainties? Uncertainties in the rate constant and mechanism for OH + isoprene? What is the ultimate impact of the results reported? Are there regimes for OH reactivity for which turbulence/air mass segregation is more or less significant (i.e. are there ranges of OH reactivity for which the effects can/cannot be neglected)?

Minor comments are listed below:

Page 1, lines 6,7 & elsewhere: Please clarify the comparisons being made and the meaning and relevance of 'resp' throughout the manuscript.

Page 1, line 12: Insert 'the' in 'during daytime'.

Page 1, line 13: Remove 'the' in 'by the ozone photolysis'.

Page 1, line 16: Change 'measured reactivity' and 'calculated reactivity' to 'measured reactivities' and 'calculated reactivities'.

Page 1, line 17: Change 'reaction constants' to 'reaction rate constants'.

Page 1, line 18 onwards: Please make the distinction between calculated reactivity (from observed concentrations of OH sinks) and modelled OH reactivity (which includes concentrations of intermediates produced in the oxidation of the observed sinks), and clarify whether the studies referred to include any model intermediates.

Page 2, lines 7-9: Provide some references and further details to the statements made.
Page 2, line 10: 'lox' to 'low'. Please include the definitions of NOx and HOx, and state the values for 'low' NOx.

Page 2, line 13: Subscript in HO2.

Page 3, line 2: 'asses' to 'assess'.

Page 3, line 25: Change 'whose' to 'in which' and 'in the spin-up' to 'of the spin-up'.

Page 3, line 33: State the locations of the observation sites.

Page 4, line 13: 'ozone gaseous precursors' to 'gaseous ozone precursors'.

Page 4, line 16: What is the justification for choosing this particular flight? Can the flight track be provided?

Page 4, line 18: Why not for NO?

Page 4, line 29: Please quantify 'very low'.

Page 5, line 5: Is ALD2 C>2 or C $\geq$ 2?

Page 5, line 15-18: What is the reason for the difference in the threshold values and why is the value of 1 taken as the threshold in this study?

Page 5, line 21: 'specie' to 'species'.

Page 5, line 29: 'ratios' to 'ratio'.

Page 6, line 8: Italicise 'i' in 'i-th'.

Page 6, line 9: 'in a similar way than' to 'in a similar way to'.

Page 6, line 21: What is the impact of this error and its neglect in previous studies?

Page 7, line 22: Please quantify 'slightly overestimates'.

Page 9, line 1: Formaldehyde is not C>2.
Page 9, line 5: Subscript in HNO4.

Page 9, line 13: 'mn' to 'min'.

Page 9, line 21 and page 10, line 20: Change 'chemical equilibrium' to 'steady state'.

Page 10, lines 4&5: Change 'lower' to 'shorter' and 'low' to 'short'.

Page 11, lines 8-11: Please clarify the meaning here. What, specifically, is implied about the reaction(s)?

Page 11, line 10: 'reactants' to 'reactant'.

Page 11, line 12: Consider changing the section title, which 'environment' does this refer to? Would 'surroudings' be a better description?

Page 11, lines 24-25: Quantify the budgets.

Page 11, line 31: 'photochemistry' to 'photochemical'.

Page 12, line 4: Change 'is maximum' to 'is at a maximum'.

Page 12, lines 11 and 21: Can the errors be shown?

Page 12, line 19: 'reactants' to 'reactant'.

Page 13, line 3: Please provide details of the 'simple chemistry'.

Page 13, line 35: Please re-word 'remains not negligible in the OH loss'. Do you mean it is significant for OH loss?

Page 14, line 15: 'imply' to 'implies'.

Page 14, line 10: 'imply' to 'implies' and 'have' to 'has'.

Page 14, line 21: 'percents' to 'percent'.

Page 15, line 17: 'HO' to 'OH'.

**ACPD**
Page 26, Fig. 1a: Why does the blue line stop at 1800 hours?

Page 30, Table 3: Subscripts in O3 and NOx.

References: There are a number of formatting issues in the references, please check thoroughly prior to publication.

---

## Referee Comment (RC2) · Anonymous Referee #2 · 9 Jan 2018

Summary

This article describes the effect of turbulence on the reactivity of OH and two key volatile organic compounds, isoprene and lumped aldehydes (where there are more than 2 carbons). The authors find that turbulence can reduce isoprene and OH reaction rates by up to 30% for a biogenic environment with low NOx, and affect reaction rates of lumped aldehyde and OH by 16% or less for an anthropogenic environment. Thus, a box model or regional and global-scale models that assume each grid box is well mixed will potentially be in error of the OH reactivity by 10% or less because of covariance of reactants caused by turbulence.

[Figure]

Over the past 5 or so years, there have been some very similar studies with similar conclusions as what was presented here (Ouwersloot et al., 2011; Kim et al., 2012; 2016; Li et al., 2016; 2017). As a consequence, the only new advance for atmospheric chemistry science is the potential error to the OH reactivity calculation in models that assume a well-mixed box. Yet this error calculation was done for a limited time and space region. In its current state, this paper is not ready to be published. Most importantly, the paper needs to emphasize its uniqueness. Otherwise it will be viewed as a confirmation of the previous studies.

Major Points

1. The paper needs to emphasize its uniqueness. The reactivity of OH should be discussed more thoroughly as it is the goal of the paper. The title of the paper, the introduction, and even the section heading (3.2 OH budget and reactivity in a convective boundary layer) say that we will learn about the convective boundary layer (CBL), yet only results at 20 m, the lowest model level, are discussed. I strongly recommend presenting OH reactivity results for the entire CBL adding a figure showing the error associated with OH reactivity when turbulence effects are considered, and stating the implications of this error on atmospheric chemistry in general. For this last part, it would be good to learn, besides the point that turbulence could explain the missing OH reactivity sink found in the Amazon, what the expectation of turbulence effects on OH reactivity is for the atmospheric oxidative capacity. Making these changes would very much align the paper's content with its title.

2. One other unique aspect of this investigation is the selected case study, which is a region in tropical West Africa that experiences stratus clouds in the morning subsequently breaking up to form cumulus clouds in the afternoon. Examining segregation of reactants with low-level stratus cloud has not been reported (to my knowledge). This aspect could be emphasized more by including the cloud fraction diurnal profile along with the other variables plotted, and address the question of what impact the cloud has on the intensity of segregation and OH reactivity. The effect is most likely limited

to the effect on heat and moisture fluxes for the simulations shown, unless effects of cloud scattering on photolysis rates are included in Meso-NH. The authors could also speculate what effect the clouds have on isoprene emissions based on transmittance (cloud optical depth) and temperature effects.

Specific Comments

1. In general, the LES results show that the intensity of segregation is <10%. The 30% segregation that is stated in the abstract and conclusions is overstated, as this high value occurs for a small-time period (∼30 minutes) and region (top of boundary layer) of the model domain where isoprene is likely quite small.

2. Although this paper does a good job of citing previous studies, I was surprised that Krol et al. (2000) was not cited. Krol et al. (2000) is one of the first to examine segregation effects on VOC + OH reaction rates.

3. Page 2, line 33. Clouds scatter radiation resulting in regions of less solar radiation and more solar radiation.

4. Page 3, lines 29-30. Could the definition of the thermals be presented in a little more detail? My understanding is that the definition is also based on vertical velocity greater than zero, the standard deviation of the decaying tracer, and for when cloud exists, liquid water content. In addition, I was wondering if radioactive-decay tracer is the correct terminology, as it could just be called a first-order decay tracer.

5. Page 4, section 2.2. How does the ReLACS 3.0 chemistry mechanism compare to that used by Kim et al. (2012)? Just based on number of chemical species, they may be quite comparable, making the statement on p. 3, line 4 inappropriate. A key part is that they both include reactions that allow peroxy radicals to produce hydroxyl radical limiting the segregation effects.

6. Page 5, section 2.3. How is the convective velocity calculated? There are different methods such that characterizing the turbulence is important.

7. Page 6. In comparing equation 4 to equation 9, should ROH be in the denominator of equation 9?

8. Page 7, lines 10-15, lines 22-24. It would be nice to see the observations overlaid on the figures displaying the model results.

9. Page 8, Is UTC same as local time (LT)? This information would be helpful when examining the diurnal profiles.

10. Page 9, lines 4-6. I am surprised production of HNO3 was not discussed. It seems a more likely outcome at temperatures near 300 K (as PAN thermally decomposes).

11. Page 10, section 3.1.2. It would be useful to show via figures why the intensity of segregation is negative or positive and why it changes with time as part of the discussion of Figure 6 on page 10. Examples of the isoprene and OH anomalies in the context of the OH production would help quantify and explain the results.

12. Page 10, lines 27-33. It is interesting that there is a positive covariance of ALD2 and OH at the top of the boundary layer from 0700 to 1230 UTC, which is during the time period of the stratus cloud (Figure 1 shows cloud fraction > 0.5). How much of this positive covariance region coincide with the clouds? How might the cloud environment affect ALD2-OH covariance (e.g. high relative humidity, scattering of photolysis rates, etc.). How much would the results differ if aqueous chemistry (or just partitioning between gas and aqueous phases) were represented in the model? Finally, why is there a positive covariance for ALD2 and OH during morning at CBL top, yet a negative covariance for isoprene and OH?

13. Page 11, section 3.2.1. It may make more sense to calculate the OH budget for several model layers to reduce effects of the emissions being injected into the lowest model layer. The lowest model layer is subject to subgrid effects (e.g., Vinuesa and Vila-Guerau, 2003; not only subgrid-scale TKE but also subgrid-scale chemistry) and therefore more uncertainties can arise in these budget calculations.

[Figure]

14. Page 12, lines 11-25. It would be good to see a figure showing the error in the OH reactivity due to turbulence effects.

15. Page 12, line 24. Since segregation happens only for compounds that have chemical lifetimes similar to the turbulence time scale, it seems unlikely that there is a compensating factor described in this line (unless isoprene + OH segregation compensates). Could the authors provide support for this statement?

16. Page 13. I applaud the authors for discussing several previous studies in the context of their results. However, I found that there was often no explanation of why there are differences between this study and the previous work. For example, p. 13, line 1, why are the current segregation values higher than Kim et al. (2016)? Line 8, how are the boundary layer dynamics different between this study and Li et al. (2016)? Lines 20-28, do the previous studies report the same dominant OH production and loss reactions as the current study? If they are different why does that happen (e.g. different emissions causes more VOCs in one study or the other)?

15. Page 15, lines 11-13. This discussion states a 5-15% missing fraction of OH reactivity reported by Nölscher et al. (2016) are similar in magnitude as that caused by turbulence effects on OH reactivity. Although I agree that the assumption of the well-mixed "box" is an issue to be considered for studies like Nölscher et al. (2016), I wonder if uncertainties in the measurements and reaction rate constants are also sufficient to explain discrepancies between model and observation analyses of OH reactivity.

16. Page 15, lines 14-20. The Li et al., 2017 study addresses segregation of reactants with aqueous phase chemistry included.

Technical Comments

1. The paper needs to be proofread carefully and completely to improve the English. There are many places that could be improved, which should have been done as part of the Quick Review Process.

2. Could "resp." be spelled out?

3. P. 12, lines 27-30, The first paragraph should cite other papers such as Krol et al., 2000; Ouwersloot et al., 2011; Kim et al., 2012; 2016; Li et al., 2016; 2017.

4. P. 18, There are two Hansen et al. (2015) references that appear to be the same.

5. Table 1. Instead of having the reader find the ReLACS3 paper, it would be helpful to explain the names of the lumped species.

6. Figure 2, I find the title of each plot to be useful. Yet in this case I do not understand "MRC". It should be defined or rewritten into something meaningful.

7. Figure 4 caption should be improved. That is, NO, NO2 should be added to a-b) description and OH, O3 should be added to c-d) description. The ALD2 line needs a scale.

8. Figure 5. What are the black lines contouring? I think thermals, but it does not say in the caption. The vertical profile lines need to be thicker.

9. Figure 6. I assume the results plotted must be an average in space, and needs to be stated in the caption.

10. Figure 8 shows results for OH reactivity at an altitude of 20 m (the lowest model level) for "updrafts", "updrafts-free", and all horizontal grid points. I think it would be better to characterize these lines as thermals and non-thermals as vertical velocity is often very close to zero at/near the surface.

References

Kim, S.-W., M. C. Barth, and M. Trainer, Influence of fair-weather cumulus clouds on isoprene chemistry, J. Geophys. Res., 117, D10302, doi:10.1029/2011JD017099., 2012. Kim S.-W., M. C. Barth, and M. Trainer, Impact of turbulent mixing on isoprene chemistry, Geophys. Res. Lett., 43, 7701–7708, doi:10.1002/2016GL069752, 2016.

[Figure]

Krol, M. C., M. J. Molemaker, and J. Vila-Guerau de Arellano, Effects of turbulence and heterogeneous emissions in photochemically active species in the convective boundary layer, J. Geophys. Res., 105, 6871–6884, doi:10.1029/1999JD900958. 2000.

Li, Y., M. C. Barth, G. Chen, E. G. Patton, S.-W. Kim, A. Wisthaler, T. Mikoviny, A. Fried, R. Clark, and A. L. Steiner, Large-eddy simulation of biogenic VOC chemistry during the DISCOVER-AQ 2011 campaign, J. Geophys. Res., 121, 8083–8105, doi:10.1002/2016JD024942, 2016.

Li, Y., Barth, M. C., Patton, E. G., and Steiner, A. L. Impact of in-cloud aqueous processes on the chemistry and transport of biogenic volatile organic compounds. J. Geophys. Res., 122. doi.org/10.1002/2017JD026688, 2017.

Nölscher, A. C., Yañez-Serrano, A. M., Wolff, S., de Araujo, A. C., Lavric, J. V., Kesselmeier, J., and Williams, J.: Unexpected seasonality in quantity and composition of Amazon rainforest air reactivity, Nature Communications, 7, 10 383, doi.org/10.1038/ncomms10383, 2016.

Ouwersloot, H. G., J. Vilà-Guerau de Arellano, C. C. van Heerwaarden, L. N. Ganzeveld, M. C. Krol, and J. Lelieveld (2011), On the segregation of chemical species in a clear boundary layer over heterogeneous land surfaces, Atmos. Chem. Phys., 11(20), 10,681–10,704, doi:10.5194/acp-11-10681-2011, 2011.

Vinuesa, J-F and J. Vilà-Guerau De Arellano, Fluxes and (co-)variances of reacting scalars in the convective boundary layer, Tellus, Ser. B, 55(4), 935–949, doi:10.1046/j.1435-6935.2003.00073.x, 2003.
* * *

---

## Author Comment (AC1) · 20 Mar 2018

Reply to the Anonymous Referee #1: We thank Referee #1 for his/her suggestions and valuable comments, which helped improving the manuscript. Comments are addressed point by point below. Extract of the manuscript are indicated in quotation marks.

Major comments:

Comment 1: The authors present a study of the impact of turbulent mixing and segregation of OH radicals and OH sinks on modelled reaction rates and OH reactivity. A

number of simulations are presented for various cases, including regions dominated by biogenic and anthropogenic emissions. However, the paper is lacking an overall summary and conclusion on the impact of these effects on model simulations for [OH] and OH reactivity.

Reply 1: The conclusion has been rewritten in order to highlight both a summary of the results and the final conclusions of our study on the impacts of turbulent mixing and segregation on OH reaction rates and reactivity (please see Reply 3 for text modification). The abstract has been also rewritten to take into account these modifications: "The hydroxyl radical (OH) is a highly reactive species and plays a key role in the oxidative capacity of the atmosphere. We explore the potential impact of a convective boundary layer on reconciling the calculation-measurement differences for OH reactivity (the inverse of OH lifetime) attributable to the segregation of OH and its reactants by thermals and the resulting modification of averaged reaction rates. The Large-Eddy simulation version of the Meso-NH model is used, coupled on-line with a detailed chemistry mechanism to simulate two contrasted biogenic and urban chemical regimes. In both environments, the top of the boundary layer is the region with the highest calculated segregation intensities but with the opposite sign. In the biogenic environment, the inhomogeneous mixing of isoprene and OH leads to a maximum decrease of 30% of the mean reaction rate in this zone. In the anthropogenic case, the effective rate constant for OH reacting with aldehydes is 16% higher than the averaged value. OH reactivity is always higher by 15 to 40% inside thermals in comparison to their surroundings as a function of the chemical environment and time of the day. Since thermals occupy a small fraction of the simulated domain, the impact of turbulent motions on domain-averaged total OH reactivity reaches a maximum decrease of 9% for the biogenic case and a maximum increase of 5% for the anthropogenic case. Accounting for the segregation of air masses by turbulent motions in regional and global models may increase OH reactivity in urban environments but lower OH reactivity in biogenic environments. In both cases, segregation alone is insufficient for resolving the underestimation between observed and modeled OH reactivity."
Comment 2: The introduction provides an overview of some previous OH reactivity measurements and studies investigating the impacts of air mass segregation. However, the studies investigating air mass segregation are typically concerned more with simulations of OH radical concentrations rather than OH reactivity. Are measurements of OH reactivity typically made on a timescale sufficiently rapid for the impacts of turbulence and segregation to be investigated?

Reply 2: The time response of current OH reactivity measurement techniques is not sufficient to directly assess the impact of turbulence. This highlights the interest of such an LES study, which is able to simulate this effect. The paper is built in two stages. First, the paper focuses on the impact of turbulence on chemical reactions of representative OH reactants (isoprene, aldehydes). Second, the model is used to calculate the impact of the inhomogeneous mixing of the chemical species on OH reactivity. The following text has been added to the introduction section (lines 25-32, p.2): "One possible issue in total OH reactivity retrieval not mentioned by previous studies could lie in neglecting turbulent motions in the transport of chemical compounds in the boundary layer. Indeed, turbulence can spatially segregate or bring together chemical species, reducing or increasing the mean reaction rate and thus chemical reactivity. However, as far as we know this physical process has not been investigated in previous studies. The time response of current OH reactivity measurement techniques is not yet sufficient to directly resolve the smallest relevant turbulent spatial scales. The limitation in time resolutions range from 30 seconds for LIF-based methods (Kovacs and Brune, 2001; Sadanaga, 2004) to one minute for the CRM method (Sinha et al., 2008). Pugh et al. (2011) and Dlugi et al. (2010) used direct isoprene and OH measurements with temporal resolution of a few seconds, fast enough to estimate the segregation of the compounds."

Comment 3: The discussion and conclusion would benefit from a discussion of the overall impacts of neglecting effects of turbulence and air mass segregation. Are the differences in [OH] and OH reactivity significant? How do they compare to measure-

ment uncertainties? Uncertainties in the rate constant and mechanism for OH + isoprene? What is the ultimate impact of the results reported? Are there regimes for OH reactivity for which turbulence/air mass segregation is more or less significant (i.e. are there ranges of OH reactivity for which the effects can/cannot be neglected

Reply 3: To address this issue, several modifications have been made in the text as the following:

[revised manuscript text omitted]

Minor comments:

Comment 4: Page 1, lines 6, 7 & elsewhere: Please clarify the comparisons being made and the meaning and relevance of 'resp' throughout the manuscript.

Reply 4: We used the abbreviation resp for respectively in the first version of our manuscript. As it seems to be confused, we removed this abbreviation throughout the revised version.

Comment 5: Page 1, line 12: Insert 'the' in 'during daytime'.

Reply 5: This has been done.

Comment 6: Page 1, line 13: Remove 'the' in 'by the ozone photolysis'.

Reply 6: This has been removed.

Comment 7: Page 1, line 16: Change 'measured reactivity' and 'calculated reactivity' to 'measured reactivities' and 'calculated reactivities'.

Reply 7: The changes have been made.

Comment 8: Page 1, line 17: Change 'reaction constants' to 'reaction rate constants'.

Reply 8: The change has been made.

Comment 9: Page 1, line 18 onwards: Please make the distinction between calculated reactivity (from observed concentrations of OH sinks) and modelled OH reactivity (which includes concentrations of intermediates produced in the oxidation of the observed sinks), and clarify whether the studies referred to include any model intermediates.

Reply 9: To avoid this confusion, the text has been changed to (lines 4-11, p.2): "Measured OH reactivity in urban areas has been shown to be similar (less than 10%) to calculated OH reactivity in New York (Ren, 2003), in Houston (Mao et al., 2010) and in a controlled urban environment (Hansen et al., 2015). However, discrepancies in urban areas have been observed between measured and calculated OH reactivity in Nashville (Kovacs et al., 2003) (35% less for calculated reactivity), in Mexico (Shirley et al., 2006) (25%) and in Tokyo (Sadanaga, 2004) (25%). The differences between measured and calculated total OH reactivity are even higher in forested areas. Di Carlo (2004) found an unexplained fraction of 50% in measured OH reactivity during the PROPHET campaign in Michigan. These results are comparable to the missing part (50 to 58%) calculated from measurements made in a boreal forest in Finland (Sinha et al., 2010; Nölscher et al., 2012). Similarly, Nölscher et al. (2016) calculated an accounted fraction of measured OH reactivity close to 49% in an Amazonian rainforest."

Comment 10: Page 2, lines 7-9: Provide some references and further details to the statements made.

Reply 10: The text has been changed to (lines 12-20, p. 3):"Attempts to use numerical models to explain the missing fraction of OH reactivity have proved to be insufficient. Indeed, Edwards et al. (2013) found an underestimation of 30% of OH reactivity in a box model with a detailed chemical mechanism for the OP3 project. In the PRIDE-

PRD campaign, Lou et al. (2010) found discrepancies of +/- 10% between the results of the OH reactivity model and the measurements, also using a box model. Similarly, Mogensen et al. (2011) used a column model to elucidate the missing part of OH reactivity, but only explained 30 to 50% of the OH reactivity measured over a forest in Finland. Chatani et al. (2009) used a three-dimensional model with coarse resolution to fill the gap in OH reactivity but 40% of the measured OH sinks remained unexplained. The difficulty of getting models to represent OH reactivity could be partly due to as yet non-discovered OH reaction pathways, and which are therefore not implemented in atmospheric models."

Comment 11: Page 2, line 10: 'lox' to 'low'. Please include the definitions of NOx and HOx, and state the values for 'low' NOx.

eply 11: This sentence has been changed to: "The OH recycling by the isoprene oxidation chain in forest environments characterized by low NOx (sum of NO and NO2) conditions (i.e. <1 ppb) was proposed to explain the uncertainties in the simulated HOx (sum of OH and HO2) budget"

Comment 12: Page 2, line 13: Subscript in HO2.

Reply 12: This has been done.

Comment 13: Page 3, line 2: 'asses' to 'assess'.

Reply 13: The change has been made.

Comment 14: Page 3, line 25: Change 'whose' to 'in which' and 'in the spin-up' to 'of the spin-up'.

Reply 14: The change has been made.

Comment 15: Page 3, line 33: State the locations of the observation sites.

Reply 15: Locations of the observations sites are now stated (lines 1-3, p.5): "The initial and forcing dynamical fields are taken from Couvreux et al. (2014) who used a

single column model to study the representation of the diurnal cycles of meteorological parameters at four observation sites inWest Africa (Agadez and Niamey in Niger, Parakou and Cotonou in Benin)."

Comment 16: Page 4, line 13: 'ozone gaseous precursors' to 'gaseous ozone precursors'.

Reply 16: The change has been made.

Comment 17: Page 4, line 16: What is the justification for choosing this particular flight? Can the flight track be provided?

Reply 17: This particular flight is described in Stone et al. (2010): it is characteristic of the boundary layer of a tropical forest in West Africa in monsoon period. More details are now given in the text (lines 18-21, p.5): "For both simulations, the initial vertical profiles of the main primary chemical species are taken from airborne measurements made during the B235 flight of the AMMA campaign performed by the BAE-146 aircraft (Table 1). This particular flight gives access to measurements performed in the boundary layer over a tropical forest in the north of Benin (10.13oN, 2.69oE) during the early afternoon (Stone et al., 2010). "

Comment 18: Page 4, line 18: Why not for NO?

Reply 18: Biogenic emissions for NO are not available in the MEGAN-MACC inventory. The only inventory given this emission is the global GEIAv1 inventory. However, available measurements of NO biogenic emissions from soil during the AMMA campaign show that biogenic fluxes for NO over West Africa from the GEIAv1 inventory are underestimated. This point is now better explained in the revised version (lines 22-27, p.5): "Biogenic emissions (Table 2) are taken from the MEGAN-MACC (Model of Emissions of Gases and Aerosols from Nature -Monitoring Atmospheric Composition and Climate) inventory (Sindelarova et al., 2014) except for NO, which is not available in this inventory. Biogenic NOx emissions from the GEIAv1 (Global Emission InitiAtive) inventory (Yienger and Levy, 1995) proved to be too low for the region studied in comparison to estimations based on airborne measurements during the AMMA campaign (Stewart et al., 2008; Delon et al., 2010). Therefore, a maximum value of 10 ng.N.m-2.s-1 was set for nitrogen oxide emissions from soils in the simulation."

Comment 19: Page 4, line 29: Please quantify 'very low'.

Reply 19: The text was modified as (line 33, p.5 to line 1, p.6): "This is in agreement with Guenther et al. (1991) who showed that isoprene emissions are thought to be null when the Photosynthetically Active Radiation (PAR) is equal to zero and maximum when the PAR exceeds the value of 1000 $\mu$mol.m-2.s-1."

Comment 20: Page 5, line 5: Is ALD2 C>2 or C#2? Reply 20: ALD2 represent the lumped C>2 aldehydes in the chemical scheme used. This is now indicated in the revised manuscript (lines 9-10, p. 6): "For the anthropogenic case, attention is given to the lumped C>2 aldehydes (ALD2 in the chemical scheme)".

Comment 21: Page 5, line 15-18: What is the reason for the difference in the threshold values and why is the value of 1 taken as the threshold in this study?

Reply 21: Considering reaction between ozone and nitrogen oxide, Schuman (1989) noticed that turbulence influence on the reaction rate between these compounds is the highest for values of Damkholer number higher than 0.1. Later studies (e.g. Molemaker and Vilà-Guerau de Arellano, 1998; Vilà-Guerau de Arellano and Cuijpers, 2000; Vilà-Guerau de Arellano and Cuijpers, 2005) rather considered that turbulence effects on reactions rates are thought to be maximum when a threshold value of 1 is used to discriminate fast (>1) and slow (<1) chemistry. To clarify this point, the text was modified as (lines 22-27, p.6):"Schumann (1989) distinguished different chemical regimes for the reaction between nitrogen oxide and ozone and found that the impact of turbulence on this reaction rate is highest for Da > 0.1. Later studies (Molemaker and Vilà-Guerau de Arellano, 1998; Vilà-Guerau de Arellano and Cuijpers, 2000; Vilà-Guerau de Arellano et al., 2005) have shown that the impacts of turbulence on atmospheric chemistry

are expected to be maximum when Da >=1. Therefore, this value will be used in the following to discriminate slow and fast chemical reactions in the boundary layer."

Comment 22: Page 5, line 21: 'specie' to 'species'.

Reply 22: The change has been made.

Comment 23: Page 5, line 29: 'ratios' to 'ratio'.

Reply 23: The change has been made.

Comment 24: Page 6, line 8: Italicise 'i' in 'i-th'.

Reply 24: The change has been made.

Comment 25: Page 6, line 9: 'in a similar way than' to 'in a similar way to'.

Reply 25: The change has been made.

Comment 26: Page 6, line 21: What is the impact of this error and its neglect in previous studies?

Reply 26: The impact of this error and the effect of neglecting it are now discussed in the revised manuscript (lines 8-15, p.8): "The segregation intensity used to compute the mean error corresponds to the deviation from the averaged boundary layer values. This error on OH reactivity was not considered in previous numerical studies focused on identifying the missing part of OH reactivity. Indeed, using a box model or a single column model like Mogensen et al. (2011), Whalley et al. (2011) or Whalley et al. (2016) leads to neglecting the turbulent motions that could affect the redistribution of chemical species within the atmospheric boundary layer. This may imply an underestimation or an overestimation of OH reactivity as a function of the sign of EROH. If EROH is positive or negative, then the effective OH reactivity REOH is either higher or lower, respectively, than the OH reactivity ROH found by neglecting the turbulent motions. Due to the crucial aspect of the OH radical in the atmosphere, this could subsequently modify the lifetimes of gaseous OH reactants such as ozone and carbon monoxide."

Comment 27: Page 7, line 22: Please quantify 'slightly overestimates'.

Reply 27: The observations of potential temperature averaged in the lowest 500m from Gounou et al. (2012) are now added to Figure 1.b and the text has been changed as (lines 11-13, p.9): "The range of simulated virtual potential temperature (Fig. 1b) overestimates the AMMA observations in Cotonou of Gounou et al. (2012) in the lowest 500m, as shown by Couvreux et al. (2014). At 0600 UTC, the model has a cold bias of -2K turning throughout the simulation to a simulated potential temperature overestimated by +2K at the end of the simulation."

Comment 28: Page 9, line 1: Formaldehyde is not C>2.

Reply 28: The text has been changed to "no aldehydes"

Comment 29: Page 9, line 5: Subscript in HNO4.

Reply 29: The change has been made.

Comment 30: Page 9, line 13: 'mn' to 'min'.

Reply 30: The change has been made.

Comment 31: Page 9, line 21 and page 10, line 20: Change 'chemical equilibrium' to 'steady state'.

Reply 31: The change has been made.

Comment 32: Page 10, lines 4&5: Change 'lower' to 'shorter' and 'low' to 'short'.

Reply 32: The change has been made.

Comment 33: Page 11, lines 8-11: Please clarify the meaning here. What, specifically, is implied about the reaction(s)?

Reply 33: This line refers to the modification of chemical reaction rates by turbulent motions. To clarify this point the text has been revised as (lines 2-5, p.13): "The previous part emphasized the non-uniform mixing between isoprene and OH for the biogenic

case and between OH and ALD2 for the urban case, and the modification of the reactions rates between these species. However, this feature must be taken into account for every OH reactant in order to obtain the full picture of total OH reactivity and gain insight into how the Meso-NH model computes the OH budget in different chemical regimes."

Comment 34: Page 11, line 10: 'reactants' to 'reactant'.

Reply 34: The change has been made.

Comment 35: Page 11, line 12: Consider changing the section title, which 'environment' does this refer to? Would 'surroudings' be a better description?

Reply 35: Thank to reviewer for his/her suggestion, the title of the section has been changed to "OH budget in thermals versus surroundings"

Comment 36: Page 11, lines 24-25: Quantify the budgets.

Reply 36: The budget is now quantify in the revised manuscript (lines 14-19, p.13) : "The OH budget for the anthropogenic case (Fig. 7b) shows that the chemical reactivity is higher inside thermals at the surface compared to the rest of the domain. The budget is largely dominated by the production of OH by NO+HO2 (79.2% of its total source in updrafts and 71.2% of the total source in non-updrafts) and by O1D+H2O (14.4% of the total source in updrafts and 18.5% of its total source in the rest of the domain). Over the whole domain, ALD2+OH (21.6% of the total loss in updrafts and 26.0% of the total loss in the rest of the domain) is the most important sink at the surface and at 500 m, followed closely by the oxidation of carbon monoxide (17% of the destruction of OH in thermals and 18.8% in non-updrafts)."

Comment 37: Page 11, line 31: 'photochemistry' to 'photochemical'.

Reply 37: The change has been made.

Comment 38: Page 12, line 4: Change 'is maximum' to 'is at a maximum'.

Reply 38: The change has been made.

Comment 39: Page 12, lines 11 and 21: Can the errors be shown?

Reply 39: The error made on the OH reactivity is now shown for both simulations (Figure 10) and discussed in lines 25-29 p.14 and lines 3-14, p.15.

Comment 40: Page 12, line 19: 'reactants' to 'reactant'.

Reply 40: The change has been made.

Comment 41: Page 13, line 3: Please provide details of the 'simple chemistry'.

Reply 41: Details have been added as (line 32, p.15 to line 1, p.16):"Using a simple chemistry scheme of 19 reactions representing the basic reactions of O3-NOx-VOC-HOx system, Ouwersloot et al. (2011) found an almost constant value of -7% for the segregation between OH and isoprene in the boundary layer over the Amazonian forest."

Comment 42: Page 13, line 35: Please re-word 'remains not negligible in the OH loss'. Do you mean it is significant for OH loss?

Reply 42: The original text has been changed to (lines 6-7, p.17):"The formaldehyde mixing ratios were close to 2 ppbv, which explains that this is not a major sink like isoprene, but remains important for OH loss through the reaction OH+HCHO."

Comment 43: Page 14, line 5: 'imply' to 'implies'.

Reply 43: The change has been made.

Comment 44: Page 14, line 10: 'imply' to 'implies' and 'have' to 'has'.

Reply 44: The change has been made.

Comment 45: Page 14, line 21: 'percents' to 'percent'.

Reply 45: The change has been made.

Comment 46: Page 15, line 17: 'HO' to 'OH'.

Reply 46: The change has been made.

Comment 47: Page 26, Fig. 1a: Why does the blue line stop at 1800 hours?

Reply 47: It is because the tracer used for this diagnostic is emitted only from 0600 to 1800 UTC to optimize computing time. A sentence was added to clarify this point in the "Dynamics" subsection of the "Simulation assessment" section (lines 9-10, p.9): "The tracer used for this diagnostic is emitted only during the period of interest, from 0600 to 1800 UTC."

Comment 48: Page 30, Table 3: Subscripts in O3 and NOx.

Reply 48: Subscripts have been added.

Comment 49: References: There are a number of formatting issues in the references, please check thoroughly prior to publication.

Reply 49: We apologize about that. The references have been carefully checked in the revised version.

Please also note the supplement to this comment:
https://www.atmos-chem-phys-discuss.net/acp-2017-969/acp-2017-969-AC1-supplement.pdf
* * *

---

## Author Comment (AC2) · 20 Mar 2018

Reply to the Anonymous Referee #2:

We thank Referee #2 for his/her suggestions and valuable comments, which helped improving the manuscript. Comments are addressed point by point below. Extract of the manuscript are indicated in italics.

Major comments:

Comment 1: This article describes the effect of turbulence on the reactivity of OH and two key volatile organic compounds, isoprene and lumped aldehydes (where there are

more than 2 carbons). The authors find that turbulence can reduce isoprene and OH reaction rates by up to 30% for a biogenic environment with low NOx, and affect reaction rates of lumped aldehydes and OH by 16% or less for an anthropogenic environment. Thus, a box model or regional and global-scale models that assume each grid box is well mixed will potentially be in error of the OH reactivity by 10% or less because of covariance of reactants caused by turbulence. Over the past 5 or so years, there have been some very similar studies with similar conclusions as what was presented here (Ouwersloot et al., 2011; Kim et al., 2012; 2016; Li et al., 2016; 2017). As a consequence, the only new advance for atmospheric chemistry science is the potential error to the OH reactivity calculation in models that assume a well-mixed box. Yet this error calculation was done for a limited time and space region. In its current state, this paper is not ready to be published. Most importantly, the paper needs to emphasize its uniqueness. Otherwise it will be viewed as a confirmation of the previous studies.

Reply 1: The originality of this work is now highlighted in the introduction section where it is specified that this work investigate the potential role of turbulent motions in explaining the observed-calculated OH reactivity discrepancies (discussed further in Reply 2). Moreover, the peculiar dynamic situation of the case study, which consists in a transition from stratus deck breaking up to form cumulus clouds, is now added in the introduction (discussed further in Reply 3). Moreover, the differences between this study and previous works studying segregation (for example Ouwersloot et al., 2011; Kim et al., 2012; 2016; Li et al., 2016; 2017) result from differences in the chemical scheme (recycling of OH during oxidation of BVOC or not, discussed further in Reply 8) and from differences in chemical species repartition in the boundary layer. This leads to lower or higher concentrations anomalies, which induce different segregation values. As an example, higher segregation is calculated in our study than in Kim et al. (2016) in the cloudy layer due to the OH vertical profiles whose gradients are sharper, which induces higher OH anomalies. Regarding the positive values of segregation simulated in the afternoon, the differences with Kim et al. (2016) might lie in discrepancies in chemical mechanism used. The OH recycling in ReLACS 3.0 is likely more important

than the one present in Mozart v2.2 used by Kim et al. (2016).

To address this point, the text was changed to (line 3-17, p.16): "In the biogenic case in this study, the negative segregation of a few percent in the middle of the boundary layer is reproduced. As in this case, higher segregation values were simulated with altitude in Kim et al. (2016), especially in the cloudy layer. However, segregation computed in the cloudy layer in Kim et al. (2016) was equal to -0.1, a value lower to that computed in the biogenic case of the present study. The discrepancies in this study and Kim et al. (2016) are likely due to the vertical OH profiles. In the study by Kim et al. (2016), OH concentrations increased linearly with altitude. This implies lower OH covariances for ascending air parcels and thus lower segregation values (Eq. 2). On the contrary, in the present study, OH is relatively homogeneous in the boundary layer and a strong gradient is present only at the top of the boundary layer. This induces high covariances for OH concentrations inside air transported by thermals at the top of the boundary layer, implying higher segregation values. As segregation computed by Li et al. (2016) is not available above 1000m height, a direct comparison with results regarding the cloudy layer is not possible with the results from the biogenic case. The positive values of segregation simulated in the afternoon (Fig. 6a) in the biogenic case are not reproduced in other studies and might be the result of efficient OH recycling in ReLACS 3.0, initiated in particular inside the thermals due to peroxy radicals formed by isoprene oxidation. Indeed, this recycling is either absent or indirect in previous works like in the mechanism used by Kim et al. (2016) that produces only HO2 from peroxy radicals, which may explain the discrepancies in OH covariances."

Comment 2: The paper needs to emphasize its uniqueness. The reactivity of OH should be discussed more thoroughly as it is the goal of the paper. The title of the paper, the introduction, and even the section heading (3.2 OH budget and reactivity in a convective boundary layer) say that we will learn about the convective boundary layer (CBL), yet only results at 20 m, the lowest model level, are discussed. I strongly recommend presenting OH reactivity results for the entire CBL adding a figure showing the

error associated with OH reactivity when turbulence effects are considered, and stating the implications of this error on atmospheric chemistry in general. For this last part, it would be good to learn, besides the point that turbulence could explain the missing OH reactivity sink found in the Amazon, what the expectation of turbulence effects on OH reactivity is for the atmospheric oxidative capacity. Making these changes would very much align the paper's content with its title.

Reply 2: The introduction section has been modified to highlight the originality of this work, especially the role of turbulence on the discrepancies in OH reactivity mentioned in the literature:

- (lines 25-32, p2): "One possible issue in total OH reactivity retrieval not mentioned by previous studies could lie in neglecting turbulent motions in the transport of chemical compounds in the boundary layer. Indeed, turbulence can spatially segregate or bring together chemical species, reducing or increasing the mean reaction rate and thus chemical reactivity. However, as far as we know this physical process has not been investigated in previous studies. The time response of current OH reactivity measurement techniques is not yet sufficient to directly resolve the smallest relevant turbulent spatial scales. The limitation in time resolutions range from 30 seconds for LIF-based methods (Kovacs and Brune, 2001; Sadanaga, 2004) to one minute for the CRM method (Sinha et al., 2008). Pugh et al. (2011) and Dlugi et al. (2010) used direct isoprene and OH measurements with temporal resolution of a few seconds, fast enough to estimate the segregation of the compounds."

- (lines 5-8, p4): "The goal of this work is to evaluate the role of thermals on OH reactivity in the framework of a convective boundary layer with contrasted chemical environments in southern West Africa. It focuses in particular on investigating turbulence as a possible explanation of the discrepancies between calculated-measured OH reactivities mentioned in the literature. Two contrasted chemical regimes represented by a detailed chemical scheme are studied by using Large-Eddy Simulations."

[Figure]

A figure was also added in the revised manuscript to show the diurnal evolution of the error associated with OH reactivity when turbulence effects are considered (Figure 10). The expectations for atmospheric chemistry and the oxidative capacity are discussed in the conclusion section (line 29, p.19 to line 9, p.20): "The overall impact of turbulence on OH concentrations and reactivity at the domain scale differs depending on the chemical environment considered. In a biogenic environment with low OH mixing ratios varying from 0.18 to 0.24 ppt, turbulent structures had little impact on the redistribution of OH in the boundary layer. This was due to an efficient OH recycling initiated by peroxy radicals formed by BVOC oxidation. In the anthropogenic case, OH mixing ratios ranged from 0.26 to 0.50 ppt. The turbulence significantly impacted the spatial distribution of OH and its precursors in the boundary layer, with higher mixing ratios in thermals. The mean relative error on domain-averaged OH reactivity revealed that effective OH reactivity (taking into account segregation by turbulent motions) in the biogenic case was up to 9% below the OH reactivity calculated based on averaged boundary layer mixing ratios. Accounting for inhomogeneous mixing between OH and its reactants (primarily isoprene) in a regional or global model could lower the calculated OH reactivity and increase the discrepancies with observed OH reactivities. In the urban environment, the mean relative error was slightly positive which means that air mass segregation by turbulence increases OH reactivity. Considering the effect of turbulent motions could reduce the gap between modelled and observed OH reactivity. However, segregation alone is unlikely to resolve the underestimation between observed and modeled OH reactivity. This study addressed the impact of moist thermals on the oxidative capacity of the atmosphere on two contrasted chemical"

Comment 3: One other unique aspect of this investigation is the selected case study, which is a region in tropical West Africa that experiences stratus clouds in the morning subsequently breaking up to form cumulus clouds in the afternoon. Examining segregation of reactants with low-level stratus cloud has not been reported (to my knowledge). This aspect could be emphasized more by including the cloud fraction diurnal profile along with the other variables plotted, and address the question of what

impact the cloud has on the intensity of segregation and OH reactivity. The effect is most likely limited to the effect on heat and moisture fluxes for the simulations shown, unless effects of cloud scattering on photolysis rates are included in Meso-NH. The authors could also speculate what effect the clouds have on isoprene emissions based on transmittance (cloud optical depth) and temperature effects.

Reply 3: The originality of this study looking at the segregation in an environment characterized by stratus cloud deck is now emphasized in the introduction. The paragraph presenting the cloud characteristics previously in the section 2.4.1 has been moved to the introduction section to complete this information (lines 21-29, p.3): "Cloud cover over West Africa is an important feature of the African monsoon but is poorly represented by global models (Knippertz et al., 2011; Hannak et al., 2017). This could lead to overly low simulated clouds and overly high incoming radiation at the surface, implying excessively high diurnal temperature and relative humidity cycles over this region. The nocturnal low-level stratus was studied during the monsoon period at Parakou (Benin) by Schrage et al. (2007) with radiosondes. The authors found that turbulent processes are responsible for cloudy nights whereas clear nights are associated with a nocturnal inversion leading to the decoupling of the surface and lower atmosphere. Schrage and Fink (2012) investigated nighttime cloud formation. They observed that the presence of a nighttime low-level jet induces the shear-driven vertical mixing of moisture accumulated near the surface. This leads to stratus formation whose cover is likely to persist until the early afternoon when it breaks up to form cumulus clouds (Schrage et al., 2007; Schrage and Fink, 2012). However, studying the impact of this specific cloudy environment on the turbulent transport of chemical species in tropical West Africa has not been reported."

The diurnal variation of cloud fraction is now added to the segregation plots. The possible cloud effects (thermodynamic fields and photolysis rates) on segregation are now discussed in the discussion section, as well as the possible impact on isoprene emissions. Finally, speculations are made on the impact of aqueous-phase chemistry

on segregation of chemical compounds in the boundary layer. These specific points and the modification of manuscript are presented in Reply 15.

Specific Comments:

Comment 4: In general, the LES results show that the intensity of segregation is <10%. The 30% segregation that is stated in the abstract and conclusions is overstated, as this high value occurs for a small-time period (30 minutes) and region (top of boundary layer) of the model domain where isoprene is likely quite small.

Reply 4: The abstract and conclusions have been corrected. The revised abstract now stated: "The hydroxyl radical (OH) is a highly reactive species and plays a key role in the oxidative capacity of the atmosphere. We explore the potential impact of a convective boundary layer on reconciling the calculation-measurement differences for OH reactivity (the inverse of OH lifetime) attributable to the segregation of OH and its reactants by thermals and the resulting modification of averaged reaction rates. The Large-Eddy simulation version of the Meso-NH model is used, coupled on-line with a detailed chemistry mechanism to simulate two contrasted biogenic and urban chemical regimes. In both environments, the top of the boundary layer is the region with the highest calculated segregation intensities but with the opposite sign. In the biogenic environment, the inhomogeneous mixing of isoprene and OH leads to a maximum decrease of 30% of the mean reaction rate in this zone. In the anthropogenic case, the effective rate constant for OH reacting with aldehydes is 16% higher than the averaged value. OH reactivity is always higher by 15 to 40% inside thermals in comparison to their surroundings as a function of the chemical environment and time of the day. Since thermals occupy a small fraction of the simulated domain, the impact of turbulent motions on domain-averaged total OH reactivity reaches a maximum decrease of 9% for the biogenic case and a maximum increase of 5% for the anthropogenic case. Accounting for the segregation of air masses by turbulent motions in regional and global models may increase OH reactivity in urban environments but lower OH reactivity in biogenic environments. In both cases, segregation alone is insufficient for resolving

the underestimation between observed and modeled OH reactivity."

Comment 5: Although this paper does a good job of citing previous studies, I was surprised that Krol et al. (2000) was not cited. Krol et al. (2000) is one of the first to examine segregation effects on VOC + OH reaction rates.

Reply 5: We apologize for this missing; the pioneering work of Krol et al. (2000) is now added to the introduction section.

Comment 6: Page 2, line 33. Clouds scatter radiation resulting in regions of less solar radiation and more solar radiation.

Reply 6: "Decrease" was changed to "modify" in the sentence (lines 18-19, p.3) : "They also modify incoming solar radiation, which in turn disturbs photolysis reactions and alters the emissions of biogenic compounds, such as isoprene."

Comment 7: Page 3, lines 29-30. Could the definition of the thermals be presented in a little more detail? My understanding is that the definition is also based on vertical velocity greater than zero, the standard deviation of the decaying tracer, and for when cloud exists, liquid water content. In addition, I was wondering if radioactive-decay tracer is the correct terminology, as it could just be called a first-order decay tracer.

Reply 7: We use the terminology of radioactive-decay tracer because it is how it was introduced by Couvreux et al. (2010). However, we agree that first-order decay may be a more simple terminology. Moreover, the definition of thermals is now presented in more details (lines 28-33, p.4): "The thermals are identified by the conditional sampling method implemented in the model by Couvreux et al. (2010), which relies on a first-order decay passive tracer mixing ratio emitted with a constant flux at the surface. In brief, in order to be considered as thermals, air parcels at a given altitude z must satisfy simultaneous conditions such as a positive vertical velocity anomaly w'>0 and tracer anomalies sv'(z) greater than the standard deviation of the tracer concentration $\sigma$sv(z) and a minimum threshold $\sigma$min(z)=(0.05/z). . In the cloud layer, a supplementary
condition is that the grid box has to be cloudy."

Comment 8: Page 4, section 2.2. How does the ReLACS 3.0 chemistry mechanism compare to that used by Kim et al. (2012)? Just based on number of chemical species, they may be quite comparable, making the statement on p. 3, line 4 inappropriate. A key part is that they both include reactions that allow peroxy radicals to produce hydroxyl radical limiting the segregation effects.

Reply 8: We agree with this statement and modified the text as (line 31, p.3 to line 4, p.4): "High resolution simulations which explicitly resolve the turbulent and convective advection terms were conducted (Vilà-Guerau de Arellano and Cuijpers, 2000; Vilà-Guerau de Arellano et al., 2005; Ouwersloot et al., 2011; Kim et al., 2012, 2016) to assess the impacts of clouds and the convective boundary layer on the mixing of chemical compounds. However, previous numerical studies on the impact of the turbulent mixing of chemical compounds mainly used relatively simple or only slightly more detailed chemical schemes (e.g., Vilà-Guerau de Arellano and Cuijpers (2000),Vilà-Guerau de Arellano et al. (2005) and Ouwersloot et al. (2011)), resulting in possible limitations in the representation of the atmospheric chemistry. However, more recent studies by Kim et al. (2012) and Kim et al. (2016) used a more detailed chemical scheme derived from Mozart v2.2, allowing the formation of OH radicals initiated by peroxy radicals. This limits the spatial heterogeneities caused by the reactions consuming the OH radical."

Comment 9: Page 5, section 2.3. How is the convective velocity calculated? There are different methods such that characterizing the turbulence is important. Reply 9: The way how the convective velocity is calculated is now described in the "Metrics subsection" (lines 21-23, p.6) : "The convective velocity is computed with g, the standard acceleration due to gravity, and Ï', the potential temperature, according to the relation where stands for the buoyancy flux at the surface."

Comment 10: Page 6. In comparing equation 4 to equation 9, should ROH be in the

denominator of equation 9?

Reply 10: ROH is in the denominator of equation 9 to normalize the estimated error made on the OH reactivity. The construction of equation 9 is intended to be similar to the one used for the segregation coefficient.

Comment 11: Page 7, lines 10-15, lines 22-24. It would be nice to see the observations overlaid on the figures displaying the model results.

Reply 11: Observations are now overlaid to the figures of boundary layer height and vertical profiles of potential temperature (Figures 1.a and 1.b). The boundary-layer height is in agreement with the observations but the models predicts a larger potential temperature diurnal cycle than observed which is similar to the results obtained with a single column models (Couvreux et al., 2014) or Numerical Weather Prediction models suggesting that at least part of those differences are due to the definition of the large-scale forcings.

Comment 12: Page 8, Is UTC same as local time (LT)? This information would be helpful when examining the diurnal profiles.

Reply 12: The local time (LT) is equal to UTC +1. This information is now given in the manuscript (lines 25-16, p.4): "The results are shown only for the third day from 0600 to 1700 UTC (LT=UTC+1), when the convective boundary-layer is well developed"

Comment 13: Page 9, lines 4-6. I am surprised production of HNO3 was not discussed. It seems a more likely outcome at temperatures near 300 K (as PAN thermally decomposes).

Reply 13: HNO3 has now been explicitly mentioned in the list of reservoir species of NO2. Besides, its production (by NO2+OH) is taken into account in the percentages of the net conversion of NO2. In the biogenic case, the low NO2 mixing ratios induce a too low production of HNO3 in the boundary layer. In the anthropogenic case, the production of HNO3 through NO2+OH reaction accounts for almost 15% of the instantaneous

destruction of NO2 at 1200 UTC near the surface. In the meantime, HNO3+OH →
NO3 +H2O and the subsequent photolysis of NO3 yields only a negligible amount of
NO2. Moreover, in the anthropogenic case, PAN is the most important reservoir of NO2
because of the efficient oxidation of carbonyls species, which produced peroxyles rad-
icals, precursors of PAN. Even if PANs thermally decompose, they accumulate in the
boundary layer, resulting in a net sink for NO2 for both cases as shown in the following
plots. (Please see figures 1,2,3 and 4 displaying PAN1 (peroxy pentionyl nitrate) and
PAN2 (peroxy acetyl nitrate) in both biogenic and anhtopogenic simulations)

The text was changed to (lines 19-24, p.10): "A minimum of NOx was found around
1200 UTC for both cases and can be explained by two factors. The first is dynamical
and is linked to the boundary layer. In the middle of the simulation, the boundary layer
growth was maximal and induced dilution in a larger mixing volume. The second factor
was chemical because at that instant, NO was efficiently converted into NO2. However,
NO2 chemical transformations in a reservoir such as HNO3, HNO4 or PANs are net
sinks for NO2. The chemical balance between reservoir species and NO2 represented
2.12% of the net destruction of NO2 averaged over the domain at 20 m and 1200 UTC
for the biogenic case, and 34.2% for the anthropogenic case. For both cases, the
main reservoirs of NO2 were PAN1 and PAN2. Therefore, less NO2 was available for
conversion into NO, which explains the low NO mixing ratios at midday (Fig. 4b and
d)."

Comment 14: Page 10, section 3.1.2. It would be useful to show via figures why the
intensity of segregation is negative or positive and why it changes with time as part of
the discussion of Figure 6 on page 10. Examples of the isoprene and OH anomalies in
the context of the OH production would help quantify and explain the results.

Reply 14: To clarify this point, as suggested by the reviewer, values of anomalies are
now indicated. For the biogenic case, the text was changed to (line 29, p.11 to line
24, p.12): "Negative values of segregation coefficients up to -30% are calculated at
the top of the cloudy boundary layer from 1000 to 1700 UTC which means that OH

and isoprene are partly segregated in this frontier zone. In other words, the hypothesis of a well-mixed atmosphere would lead to a 30% overestimation of the reaction rate at the frontier between the boundary layer and the free troposphere. The negative segregation (Eq. 2) means the anomalies of isoprene and OH have opposite signs, as shown in figure 5a. This is due to lower OH mixing ratios in thermals than in the environment (Fig. 5b). These results are consistent with the previous studies of Li et al. (2016); Kim et al. (2016); Ouwersloot et al. (2011) (see Discussion). In the biogenic case, isoprene anomalies in thermals are considerable from the surface (+0.48 ppbv on average at midday) to the top of the boundary layer (+0.1 ppbv on average at midday) and are thought to be always positive as OH is uniformly emitted at the ground (Fig. 5a). On the contrary, OH mixing ratios are almost constant in the boundary layer at 1200 UTC (Fig. 5b), so the magnitude of OH anomalies are expected to be low (-0.03 pptv on average at midday at the top of the boundary layer). Besides, due to its very short chemical lifetime, OH quickly reaches equilibrium with its surroundings, implying that its fluctuations are mostly due to thermals transporting air originating from different chemical environments. Thus, isoprene anomalies are thought to be the major driver of the magnitude of segregation over the boundary layer whereas changes in OH anomalies are related to changes in the segregation sign. Positive values around +5% are calculated at 700 meters starting from 1400 to 1800 UTC (Fig. 6a). The intensity of segregation becomes positive due to positive anomalies of both compounds. Due to decrease in isoprene emissions in the afternoon, OH destruction slows down, especially inside thermals. They are still active in transporting enough NO to react with HO2 to produce OH, inducing higher OH mixing ratios inside updrafts than in the surroundings (+0.02 pptv on average at 1600 UTC). Before 0900 UTC near the surface, the segregation coefficient in the anthropogenic simulation between OH and ALD2 is negative up to -8% in the lower 200m (Fig. 6b), due to the anthropogenic emission patch. As chemical equilibrium is not yet reached, more of the OH radical is destroyed through its reaction with recently emitted compounds than that which is produced (not shown). This means that OH is less concentrated inside updrafts at

that moment so its anomalies are negative near the surface. Simultaneously, positive segregations develop at the top of the boundary layer from 0700 UTC to 1230 UTC with a maximum of 16% at 1000 UTC and from 1530 to 1730 UTC. The positive segregation is related to the concomitant transport of ALD2 and precursors of OH by thermals. Moreover, the high segregation values correspond to the presence of clouds between 0.6 and 0.9 km (Fig. 2c), simultaneously with a high cloud fraction over the domain (>0.6) (Fig. 6b). This specific point is discussed in the discussion section. As ALD2 is emitted at the surface, its anomalies are high and positive inside updrafts. For example, at midday, anomalies are +0.5 ppbv on average at the surface and nearly +4 ppbv at the top of the boundary layer. However, in this case, OH anomalies are more difficult to predict due to the spatial heterogeneities of chemical emissions. Local changes in OH production and destruction explain the changes in the segregation sign throughout the simulation."

Comment 15: Page 10, lines 27-33. It is interesting that there is a positive covariance of ALD2 and OH at the top of the boundary layer from 0700 to 1230 UTC, which is during the time period of the stratus cloud (Figure 1 shows cloud fraction > 0.5). How much of this positive covariance region coincide with the clouds? How might the cloud environment affect ALD2-OH covariance (e.g. high relative humidity, scattering of photolysis rates, etc.). How much would the results differ if aqueous chemistry (or just partitioning between gas and aqueous phases) were represented in the model? Finally, why is there a positive covariance for ALD2 and OH during morning at CBL top, yet a negative covariance for isoprene and OH?

Reply 15: The presence of clouds concomitant to high segregation between ALD2 and OH is now added in the section 3.1.2. (see manuscript changes in the previous comment for this specific point). The modification of ALD2-OH covariance by the presence of clouds is now discussed in the discussion section (lines 23, p.17 to line 12, p.18): "In the anthropogenic simulation, the high values of segregation computed at the top of the boundary layer are coincident with the presence of clouds. In the absence of

aqueous-phase chemistry, clouds impacts on chemical species are dynamical and photochemical. They modify heat and moisture fluxes at their surroundings (Vilà-Guerau de Arellano et al., 2005) and thus the transport of compounds, as noted by Kim et al. (2012) who demonstrated that clouds presence could increase transport of chemicals to 1000m. In this set of simulations, the chemical impact of clouds on species involves photolysis rates as they are corrected at every time step due to the presence of clouds according to the work of Chang et al. (1987). At each point of the domain, photolysis rates are increased above clouds and decreased below them. Another effect of clouds on the atmospheric chemistry lies in isoprene emissions, as demonstrated by Kim et al. (2012). As isoprene emissions are dependent on incoming radiation and temperature near the surface, cloud shading could decrease the amount of isoprene emitted. Kim et al. (2012) showed that isoprene concentrations are decreased by 10% and OH concentrations increased by 5% in the boundary layer when isoprene emissions are reduced by up to 10%. The ultimate impact onsegregation cannot be anticipated since it corresponds to two compensating effects. Finally, clouds impact atmospheric chemistry through aqueous phase reactivity. Aqueous-phase chemistry was not considered here, nor were the exchanges between gas and aqueous phases. However, it could have an impact on soluble species mixing ratios, such as formaldehyde and H2O2 through the capture and degassing cycles of these compounds. Lelieveld and Crutzen (1990) showed a decrease in oxidative capacity of the atmosphere through aqueous-phase reactions via a significant decrease in ozone mixing ratios, but 5 also OH, formaldehyde and nitrogen oxides. However, the effects of aqueous-phase chemistry on gas-phase compound concentrations are various (Barth et al., 2003) and OH concentrations could decrease in clouds (Mauldin et al., 1997). This result was confirmed by the study of Commane et al. (2010) who found that HOx concentrations decreased in clouds. Recently, Li et al. (2017) studied segregation effects in a biogenic environment when aqueous-phase chemistry is included. They found that isoprene concentrations are increased by up to 100% while OH concentrations decreased by 18%, resulting in a maximum segregation of 55% in the cloudy layer. In the anthropogenic environment,

segregation effects are expected to be enhanced due to the decrease of OH concentrations in gaseous phase in the cloud layer, reducing the cleansing capacity of the atmosphere." In the anthropogenic case, the positive covariance for ALD2 and OH at the top of the CBL during the morning is a consequence of transport of ALD2, emitted at the surface, and OH precursors, also emitted at the surface. In the biogenic case, isoprene, emitted at the surface, is more concentrated in thermals, which means positive anomalies of concentrations over the domain. The OH radical is slightly less concentrated in updrafts (-0.03 pptv at the top of the boundary layer). It is the result of a chemical balance characterized by its destruction (chemical reactions, especially isoprene) and its production (chemical production by NO+HO2 reaction but also its recycling by peroxy radicals). This balance changes over time depending of the local chemical regime, which changes the sign of OH covariance in the boundary layer and thus affects the segregation sign. In the morning and at the top of the boundary layer, this chemical balance induces lower OH concentrations in updrafts and so negative covariance for isoprene and OH.

Comment 16: Page 11, section 3.2.1. It may make more sense to calculate the OH budget for several model layers to reduce effects of the emissions being injected into the lowest model layer. The lowest model layer is subject to subgrid effects (e.g., Vinuesa and Vila-Guerau, 2003; not only subgrid-scale TKE but also subgrid-scale chemistry) and therefore more uncertainties can arise in these budget calculations.

Reply 16: The OH budget at 20m allows a comparison between model results and measurements. Nevertheless, the text had been modified to include the uncertainties rising from the subgrid effects (lines 3-6, p.13): "This height is the first level in the model and computing the chemical budget at this height leads to uncertainties due to subgrid-scale mixing and chemistry (Vinuesa and Vilà-Guerau de Arellano, 2005). However, it makes it possible to compare the model results with the measurements in the literature."

Moreover, figures showing the OH budget at the top of the boundary layer are now

added to the manuscript with the following associated discussion in section 3.1.2 (lines 25, p.13 to line 17, p.14): "The chemical budget at 1200 m (Fig. 8), namely at the top of the boundary layer, allows the investigation of chemical reactions inside the ascending air parcel lifted by thermals and its comparison with its surroundings. For the biogenic case, the major OH reactants in the thermals have a chemical lifetime greater than the turbulence timescale. At this altitude, only species whose Damkhöler numbers are lower than 1 are present in sufficient amounts to react with the OH radical. For example, carbon monoxide (26.2% of total OH destruction in thermals and 36.6% in the surroundings) is the major sink, but also methane (11.8% of total OH loss in updrafts and 18.4% in the environment). Chemical compounds with a secondary source like formaldehyde and C>2 aldehydes (ALD2) are other important sinks at 1200m. Isoprene, the major OH reactant close to the surface, is present only in thermals at this altitude due to its reaction with OH in the ascending air parcel and consumes 11.8% of OH in thermals. OH production by NO+HO2 reaction is null inside updrafts and low in the non-updraft area (3.2% of the total OH production) due to NO destruction in updrafts. The reaction between hydroxyl radicals RO2, secondary products, with HO2 is a major OH source in thermals (49.7% of total production) but also in the surroundings (36.1% of OH production in updraft-free region). Production of OH by O1D + H2O or H2O2 photolysis are similar in magnitude in thermals and in the non-thermal areas. The production and destruction terms are higher in the thermals compared to these terms in non-updrafts due to higher concentrations of OH reactants inside the thermals, but lower than at 20m. Regarding the anthropogenic case, species whose lifetimes are higher than the turbulence timescale are major OH reactants at 1200m. Carbon monoxide contributes 22% of the OH destruction in thermals and 34.3% in the rest of the domain. As with the biogenic case, chemical compounds with a secondary source are important OH sinks like formaldehyde (11.1% in thermals and 11.0% in updraft-free regions) and ALD2 (26.4% of total OH destruction in updrafts and 11.8% in the surroundings), corresponding to the major OH destruction term at the top of the boundary layer. The OH production terms in the anthropogenic case are similar in thermals compared to the surface with a high contribution of NO + HO2 reaction (66.9% of total OH production), followed by O1D + H2O reaction (15.3%) and RO2 + HO2 (10.6%). In the rest of the domain, NO + HO2 contribution drops to 16.9% while O1D + H2O (32.8%) and RO2 + HO2 (31.1%) are major production terms. The differences between the OH reactivity at the surface and the top of the boundary layer are mainly driven by the changes in chemical mixing ratios of precursors caused by chemical reactions and consequently by their Damkhöler number, and by the secondary products formed inside the thermals."

Comment 17: Page 12, lines 11-25. It would be good to see a figure showing the error in the OH reactivity due to turbulence effects.

Reply 17: A figure showing the error in the OH reactivity had been added to the manuscript for both biogenic and anthropogenic cases with the following associated discussion in section 3.2.2 (lines 13-18, p.15): "In both cases, the occurrence and development of clouds (Fig. 6, upper panel) is concomitant with linear increases of the error made on the OH reactivity while neglecting the impact of turbulence (Fig. 10). The diurnal cycle of ERoh in each case is correlated to the development of the convective boundary layer. Firstly, a rapid change occurs during the first hours of the simulations, characterized by the occurrence of thermals and an increase in chemical emissions for the biogenic environment. Then, ERoh is relatively stable from the end of the morning to the middle of the afternoon, with a maximum value computed around 1400 UTC, corresponding to the maximum turbulent activity in the convective boundary layer." Comment 18: Page 12, line 24. Since segregation happens only for compounds that have chemical lifetimes similar to the turbulence time scale, it seems unlikely that there is a compensating factor described in this line (unless isoprene + OH segregation compensates). Could the authors provide support for this statement? Reply 18: Segregation between two compounds happens when at least one of the two compounds have chemical lifetimes lower than or equal to the turbulence time scale. In that sense, segregation might be important between OH and each of its reactants. Moreover, the

segregation used to compute the error made on the OH reactivity is now calculated from boundary layer averaged values. In other words, covariances of each compound are computed relatively to concentration averaged over the whole boundary layer. Depending on the chemical species considered and their vertical distribution, averaging these covariances to compute the segregation coefficient can either induce negative values of segregation or lower positive values, even if positive values of segregation are computed when considering vertical segregation. As an example, the segregation computed relatively to boundary layer averages for ALD2 and OH is positive during daytime but does not exhibit the large values simulated in Fig. 6, because of the average over the boundary layer. (Please see figure 5 displaying boundary layer averaged segregation between ALD2 and OH in the anthropogenic simulation) The error made on the OH reactivity is thus the result of every OH reactants segregation like CO or formaldehyde (major OH sinks in the boundary layer). (Please see figure 6 displaying boundary layer averaged segregation between CO and OH and formaldehyde and OH in the anthropogenic simulation)

To clarify this point, the text was changed to (lines 10-12, p.15): "Moreover, chemicals have either negative or positive segregation towards OH that may compensate or increase the positive values simulated for ALD2 and OH (Fig. 6b)."

Comment 19: Page 13. I applaud the authors for discussing several previous studies in the context of their results. However, I found that there was often no explanation of why there are differences between this study and the previous work. For example, p. 13, line 1, why are the current segregation values higher than Kim et al. (2016)? Line 8, how are the boundary layer dynamics different between this study and Li et al. (2016)? Lines 20-28, do the previous studies report the same dominant OH production and loss reactions as the current study? If they are different why does that happen (e.g. different emissions causes more VOCs in one study or the other)?

Reply 19: The differences with Li et al. (2016) cannot be discussed for the cloudy layer as the segregation is only computed from the surface to 1000m in Li et al. (2016).

[revised manuscript text omitted]

Comment 20: Page 15, lines 11-13. This discussion states a 5-15% missing fraction of OH reactivity reported by Nölscher et al. (2016) are similar in magnitude as that caused by turbulence effects on OH reactivity. Although I agree that the assumption of the well- mixed "box" is an issue to be considered for studies like Nölscher et al. (2016), I wonder if uncertainties in the measurements and reaction rate constants are also sufficient to explain discrepancies between model and observation analyses of OH reactivity.

Reply 20: We agree that all this factors or a combination of them could explain the

discrepancies. This was added in the discussion section (lines 34, p.18 to line 6, p.19): "Moreover, the error made on the total OH reactivity neglecting the turbulent mixing could cumulate with the uncertainties reported in the literature regarding OH reactivity techniques such LIF with a flow tube (from 10 to 15%, Kovacs and Brune (2001)), LP-LIF (from 10 to 20%, Sadanaga (2004)) and the CRM measurement method (15 to 20%, Sinha et al. (2008)). In addition, uncertainties on reaction rate constants are also present in chemical schemes, including those used by numerical models. These uncertainties on reaction rate coefficients range from 5 to 15%, as suggested by Atkinson et al. (2006). It is likely that the unaccounted fraction of OH reactivity reported in the literature may be explained 5 at least partially by a combination of the following phenomena similar in intensity: turbulence effects on chemical reactivity and uncertainties on the OH reactivity measurements and reaction rate coefficients."

Comment 21: Page 15, lines 14-20. The Li et al., 2017 study addresses segregation of reactants with aqueous phase chemistry included.

Reply 21: This publication is now present in the discussion section where clouds presence and segregation are discussed (see Reply 15).

Technical Comments

Comment 22: The paper needs to be proofread carefully and completely to improve the English. There are many places that could be improved, which should have been done as part of the Quick Review Process.

Reply 22: The revised manuscript was proofread by a native English speaker.

Comment 23: Could "resp." be spelled out ?

Reply 23: This abbreviation was not used anymore in the revised manuscript.

Comment 24: P. 12, lines 27-30, The first paragraph should cite other papers such as Krol et al.,2000; Ouwersloot et al., 2011; Kim et al., 2012; 2016; Li et al., 2016; 2017.

Reply 24: The paragraph is now modified to (lines 20-26, p.15):"The redistribution of chemical species by the boundary layer turbulence induces a different mean reaction rate between compounds when compared to a situation in which chemical species would be perfectly mixed (Krol et al. (2000), Ouwersloot et al. (2011), Kim et al. (2012), Kim et al. (2016), Li et al. (2016) and Li et al. (2017) among others). The perfectly mixed assumption used in regional and large scale atmospheric models leads to errors on the mean reaction rates between species as the turbulent mixing occurs at scales smaller than the grid length (Vinuesa and Vilà-Guerau de Arellano, 2005). This implies that the OH total reactivity has been calculated inaccurately, in turn leading to a modification in the lifetimes of the OH reactants such as ozone and carbon monoxide."

Comment 25: P. 18, There are two Hansen et al. (2015) references that appear to be the same.

Reply 25: This has been corrected.

Comment 26: Table 1. Instead of having the reader find the ReLACS3 paper, it would be helpful to explain the names of the lumped species.

Reply 26: Species names are now included in the Table 1 caption.

Comment 27: Figure 2, I find the title of each plot to be useful. Yet in this case I do not understand "MRC". It should be defined or rewritten into something meaningful.

Reply 27: "MRC" stands for cloud water mixing ratio. This is now stated in the title.

Comment 28: Figure 4 caption should be improved. That is, NO, NO2 should be added to a-b) description and OH, O3 should be added to c-d) description. The ALD2 line needs a scale. Reply 28: The caption has been corrected and ALD2 line has now a scale.

Comment 29: Figure 5. What are the black lines contouring? I think thermals, but it does not say in the caption. The vertical profile lines need to be thicker.

Reply 29: Yes, black lines are thermals. The caption has been corrected.

Comment 30: Figure 6. I assume the results plotted must be an average in space, and needs to be stated in the caption.

Reply 30: The variable plotted in Fig.6 is the segregation coefficient that is, by definition, an average in space.

Comment 31: Figure 8 shows results for OH reactivity at an altitude of 20 m (the lowest model level) for "updrafts", "updrafts-free", and all horizontal grid points. I think it would be better to characterize these lines as thermals and non-thermals as vertical velocity is often very close to zero at/near the surface.

Reply 31: We agree with the reviewer and modified the caption in that sense.

Please also note the supplement to this comment:
https://www.atmos-chem-phys-discuss.net/acp-2017-969/acp-2017-969-AC2-supplement.pdf
* * *
[Figure]

[Figure]

**Fig. 1.**

[Figure]

**Fig. 2.**

PAN1 (ppb) [0800UTC]

PAN1 (ppb) [1200UTC]

PAN1 (ppb) [1600UTC]

**Fig. 3.**

PAN2 (ppb) [0800UTC]

PAN2 (ppb) [1200UTC]

PAN2 (ppb) [1600UTC]

**Fig. 4.**

[Figure]

[Figure]

**Fig. 5.**

[Figure]

**Fig. 6.**